# Anti-HMGB1 Antibody Therapy Ameliorates Spinal Cord Ischemia–Reperfusion Injury in Rabbits

**DOI:** 10.3390/ijms26178643

**Published:** 2025-09-05

**Authors:** Genya Muraoka, Yasuhiro Fujii, Keyue Liu, Handong Qiao, Dengli Wang, Daiki Ousaka, Susumu Oozawa, Shingo Kasahara, Masahiro Nishibori

**Affiliations:** 1Department of Cardiovascular Surgery, Okayama University Graduate School of Medicine, Dentistry and Pharmaceutical Sciences, Okayama 700-8558, Japan; plsm7x1k@okayama-u.ac.jp; 2Department of Translational Research, Center for Innovative Clinical Medicine, Medical Development Field, Okayama University, Okayama 700-8558, Japan; 3Department of Pharmacology, Faculty of Medicine, Dentistry and Pharmaceutical Sciences, Okayama University, Okayama 700-8558, Japan; liukeyue@md.okayama-u.ac.jp (K.L.); cpuqiaoljjljjljj@163.com (H.Q.); dengliwang@okayama-u.ac.jp (D.W.); 4Department of Medical Technology, Faculty of Science, Okayama University of Science, Okayama 700-0005, Japan; d-ousaka@ous.ac.jp; 5Division of Medical Safety Management, Safety Management Facility, Okayama University Hospital, Okayama 700-8558, Japan; ohzawa-s@cc.okayama-u.ac.jp; 6Department of Cardiovascular Surgery, Faculty of Medicine, Dentistry and Pharmaceutical Sciences, Okayama University, Okayama 700-8558, Japan; shingok@md.okayama-u.ac.jp; 7Department of Translational Research and Drug Development, Faculty of Medicine, Dentistry and Pharmaceutical Sciences, Okayama University, Okayama 700-8558, Japan; mbori@md.okayama-u.ac.jp

**Keywords:** thoracoabdominal aortic aneurysm, spinal cord ischemia–reperfusion injury, high mobility group box 1, neuroprotection, blood–spinal cord barrier, aortic surgery

## Abstract

Spinal cord ischemia–reperfusion (SCI/R) injury remains a major clinical challenge with limited therapeutic options. High-mobility group box 1 (HMGB1), a proinflammatory mediator released during cellular stress, has been implicated in the pathogenesis of ischemia–reperfusion-induced neural damage. In this study, we investigated the neuroprotective potential of the anti-HMGB1 monoclonal antibody (mAb) in a rabbit model of SCI/R injury. Male New Zealand White rabbits were anesthetized and subjected to 11 min of abdominal aortic occlusion using a micro-bulldog clamp following heparinization. Anti-HMGB1 mAb or control IgG was administered intravenously immediately after reperfusion and again at 6 h post-reperfusion. Neurological function was assessed at 6, 24, and 48 h after reperfusion using the modified Tarlov scoring system. The rabbits were euthanized 48 h after reperfusion for spinal cord and blood sampling. Treatment with anti-HMGB1 mAb significantly improved neurological outcomes, reduced the extent of spinal cord infarction, preserved motor neuron viability, and decreased the presence of activated microglia and infiltrating neutrophils. Furthermore, it attenuated apoptosis, oxidative stress, and inflammatory responses in the spinal cord, and helped maintain the integrity of the blood–spinal cord barrier. These findings suggest that anti-HMGB1 mAb may serve as a promising therapeutic agent for SCI/R injury.

## 1. Introduction

Spinal cord ischemia (SCI), which can lead to paraplegia or paraparesis, remains one of the most severe complications following thoracoabdominal aortic aneurysm (TAAA) surgery, encompassing both conventional open surgery and thoracic endovascular repair [1,2]. Permanent lower extremity paraplegia due to SCI has been reported in 3.3% of patients undergoing treatment for TAAA, with an incidence of 4.0% following open surgery and 2.9% after endovascular repair [3]. Moreover, it has been suggested that the proportion of delayed-onset spinal cord injury among all SCI cases after TAAA repair, whether open or endovascular, may be unexpectedly high, potentially exceeding 80% [4].

Clinically, two types of paraplegia or paraparesis may develop after TAAA surgery: immediate and delayed onset [5,6]. Immediate paraplegia stems from irreversible ischemic neuronal injury to the spinal cord. Conversely, delayed-onset paraplegia is believed to result from SCI/R injury [7], which can be potentially attributed to the initial ischemic stress and subsequent molecular changes during the reperfusion phase [8,9]. Ischemia–reperfusion (I/R) injury occurs when blood supply is restored after prolonged ischemia, leading to local inflammation, increased production of reactive oxygen species (ROS), and secondary cell injury or death [10].

High mobility group box 1 (HMGB1), which was initially identified as a non-histone DNA-binding nuclear protein, is ubiquitously present in the nuclei of eukaryotic cells. Recent studies have recognized HMGB1 as a damage-associated molecular pattern molecule that triggers proinflammatory responses [11,12], and plays an important role in various central nervous system (CNS) disorders [13,14]. Our previous work demonstrated that neutralizing HMGB1 with a monoclonal antibody (mAb) significantly reduced brain infarction [15], hemorrhage [16], and traumatic brain injury [17] in rat models, primarily by protecting the blood–brain barrier (BBB) and inhibiting the inflammatory cascade [18]. Additionally, other research groups have reported that inhibiting the HMGB1-Toll-like receptor 4 (TLR4)-nuclear factor kappa-B (NF-κB) signaling pathway using HMGB1 inhibitors such as dexmedetomidine [19], ethyl pyruvate [20], and glycyrrhizin [21,22] can alleviate I/R injury in the spinal cord of rabbit models. However, the direct effects of the HMGB1 blockade on SCI/R injury have not yet been confirmed.

In this study, we established a rabbit model of delayed SCI/R injury and evaluated the efficacy of an anti-HMGB1 mAb in preventing this condition. We focused primarily on its effects on the blood–spinal cord barrier (BSCB) and inflammatory responses, further assessing its ability to ameliorate neurological dysfunction. Collectively, these investigations aim to clarify the preventive potential of anti-HMGB1 mAb as a novel neuroprotective strategy for patients at risk of spinal cord injury during TAAA repair.

## 2. Results

### 2.1. Varied Neurological Outcomes in Rabbit SCI/R Injury

To evaluate the relationship between the duration of SCI and neurological deficits, we examined various aortic clamp times ranging from 8 to 30 min in a rabbit model of SCI/R injury. As shown in Table 1, rabbits subjected to clamp times shorter than 11 min exhibited no significant neurological deficits at 6, 24, and 48 h after reperfusion. In contrast, complete paraplegia was observed as early as 6 h post-reperfusion and persisted for 48 h in rabbits subjected to clamp times exceeding 11 min. Notably, a clamp time of 11 min induced a unique pattern of delayed neurological deterioration: No deficits were evident at 6 h, but symptoms progressively worsened, with the mean modified Tarlov scores decreasing to 3 at 24 h and 1.5 at 48 h. On the basis of these findings, we inferred that an 11 min abdominal aortic clamp is the optimal model for evaluating delayed spinal cord injury caused by I/R.

### 2.2. Anti-HMGB1 mAb Improves Neurological Deficits After SCI/R Injury

Impairments in hind limb motor function after I/R injury were evaluated using the modified Tarlov score (Figure 1). In the control IgG group, clear neurological deficits developed after I/R injury and showed a progressive worsening at 24 and 48 h. Treatment with anti-HMGB1 mAb significantly ameliorated these deficits compared with the control IgG group; however, the recovery did not reach the intact level observed in the Sham group. Notably, rabbits in the Sham group exhibited no neurological deficits throughout the observation period. The anti-HMGB1 mAb and Control IgG groups showed no differences in oxygen saturation, heart rate, systolic and diastolic blood pressures, or body temperature during the procedure (Table 2). Plasma HMGB1 levels showed a progressive increase over time following the surgical procedure in both groups. No significant differences were observed between the anti-HMGB1 mAb and control IgG groups. In the Sham group, although a slight increase was observed, the values remained nearly equivalent to the preoperative level, and no substantial changes were detected (Figure 2).

All animals underwent SCI/R injury. The animals in the control IgG or anti-HMGB1 monoclonal antibody (α-HMGB1) groups were exposed to the control IgG or α-HMGB1, respectively, after an 11 min SCI/R injury. Values represent means ± SEM in control IgG group (*n* = 5) and α-HMGB1 group (*n* = 5). In the Sham group, the median is reported for two subjects. BT, body temperature; DBP, diastolic blood pressure; HMGB1, high-mobility group box 1; HR, heart rate; SBP, systolic blood pressure; SpO_2_, percutaneous oxygen saturation.

### 2.3. Histological Studies on the Effects of Anti-HMGB1 mAb

Representative coronal sections fixed 48 h after I/R injury and stained with hematoxylin and eosin are shown in Figure 3. In the anti-HMGB1 mAb group with a Tarlov score of 4, numerous viable motor neurons exhibited polygonal structures and Nissl bodies, which are characterized by basophilic staining in the cytoplasm (equivalent to the Sham group level). Conversely, as the Tarlov score decreased, the number of viable motor neurons in the anterior horn also diminished. Along with this decline, the morphology of Nissl bodies showed progressive alterations: from large, darkly stained structures with well-defined boundaries to smaller, paler bodies with indistinct margins. In the control IgG group with a Tarlov score of 1, imaging revealed extensive infarct lesions with impaired motor neurons, which were characterized by eosinophilic cytoplasmic dye and vacuolization.

Figure 4 shows the analysis of viable motor neurons in the anterior horn of the spinal cord at the L5 to L7 levels, which were presumed to be affected by reduced blood flow following abdominal aortic clamping. We first compared the number of viable motor neurons in a single transverse section of the anterior horn at the L7 level, which is the most distal and therefore expected to be the most severely affected by ischemia. Although this comparison did not reach statistical significance (*p* = 0.126), the Cliff’s delta value was +0.84, indicating a very strong effect size (Figure 4A). On the basis of this finding, we then quantified viable motor neurons in a single transverse section at the level of L5, L6, and L7 levels and analyzed the data on a per-animal basis. The anti-HMGB1 mAb group demonstrated a significantly greater total number of viable anterior horn motor neurons compared with the control IgG group (*p* < 0.05), indicating a neuroprotective effect of the anti-HMGB1 antibody (Figure 4B). Furthermore, Figure 4C illustrates the number of viable motor neurons 48 h after SCI/R injury, assessed at three spinal cord levels (L5, L6, and L7). The Sham group (*N* = 1) exhibited no neurological deficits and maintained motor neuron counts comparable to the preoperative state; however, statistical comparisons with this group were not performed because of the limited sample size. In the control IgG and anti-HMGB1 mAb groups (*N* = 5 each), no statistically significant differences were detected at any spinal cord level. Notably, in both groups, a few animals displayed exceptionally high numbers of surviving motor neurons, which could be regarded as outliers. These outliers likely accounted for the absence of statistical significance despite the apparent trend toward neuronal preservation in the anti-HMGB1 mAb group. Such cases are considered to reflect inter-individual variability in the development of collateral circulation within the spinal cord. Supporting this interpretation, outliers were less frequently observed at the L7 level compared with L5 and L6, suggesting that collateral blood supply becomes progressively less effective at more distal levels from the aortic clamping site. This phenomenon is further explained by the anatomical fact that rabbits possess fewer collateral pathways in the spinal cord than humans.

### 2.4. Anti-HMGB1 mAb Inhibits Microglia and Neutrophil Activation and HMGB1 Release

To investigate the effects of the anti-HMGB1 mAb on spinal cord inflammation, we examined intracellular HMGB1 translocation and retention, the number of microglial cells, and neutrophil infiltration by immunofluorescence staining in the anterior horn of the spinal cord following I/R injury (Figure 5). The number of cells retaining HMGB1 was reduced in the control IgG group, and this reduction was attenuated by anti-HMGB1 mAb treatment, resulting in levels comparable to those observed in the Sham group. Furthermore, the expression of Iba1, a specific marker for microglia, was elevated in the control IgG group, but decreased in the anti-HMGB1 mAb group. Representative immunofluorescence images are shown in Figure 5A, in which only a few cells exhibit double labeling of Iba1 and HMGB1. Quantification of the area occupied by anti-HMGB1-positive cells and the number of microglia revealed statistically significant differences between the control IgG and anti-HMGB1 mAb groups (Figure 5B,C). These findings suggest that the spinal cord damage induced by ischemia is attenuated by the administration of anti-HMGB1 mAb. Similar results were observed when myeloperoxidase (MPO) was stained with a fluorescent dye (Figure 6A). The number of MPO-positive neutrophils in the control IgG group was greater than that in the Sham group. After administration of the anti-HMGB1 mAb, this increase was significantly suppressed to levels comparable to those in the Sham group (Figure 6B). These findings indicated a reduction in neutrophil infiltration during I/R, which was attributed to the administration of anti-HMGB1 mAb.

### 2.5. Caspase-3 and Oxidative Stress Are Reduced by Anti-HMGB1 mAb

At 48 h after I/R, oxidative stress and neuronal apoptosis in the spinal cord were evaluated by quantifying 4-hydroxynonenal (4-HNE), a marker of lipid peroxidation, and cleaved caspase-3, a key executor of apoptosis, using Western blot densitometry (Figure 7). Representative Western blot images of each marker are also shown above the graphs. The expression of 4-HNE was significantly elevated in the control IgG group compared with the Sham group, and this increase was significantly suppressed by anti-HMGB1 mAb treatment (*p* < 0.05) (Figure 7A). Similarly, cleaved caspase-3 expression markedly increased following I/R injury but was substantially attenuated by anti-HMGB1 mAb treatment (*p* < 0.001) (Figure 7B). The Sham group (*n* = 1) is shown as a reference only and was not included in the statistical analyses. Together, these results demonstrate that anti-HMGB1 mAb attenuates both oxidative stress and apoptosis after SCI/R injury.

### 2.6. Anti-HMGB1 mAb Administration Preserves BSCB

To assess BSCB permeability in I/R injury models, we performed immunofluorescence analysis to quantify albumin leakage in the spinal cord after I/R injury. Immunofluorescence analysis revealed that albumin leakage predominantly occurred in the gray matter (Figure 8). These findings demonstrate that anti-HMGB1 mAb administration exerts protective effects against BSCB permeability following spinal I/R injury.

To further assess the detrimental impact of SCI/R injury on BSCB integrity, immunofluorescence staining was performed for key BSCB proteins, including glial fibrillary acidic protein (GFAP), a marker of astrocytes, and platelet-derived growth factor receptor β (PDGFRβ), a marker of pericytes. The staining revealed marked structural disruptions in the BSCB in the control IgG group, characterized by prominent gaps between PDGFRβ-positive pericytes and astrocytic processes, indicative of astrocytic swelling and endfoot detachment (white arrows in Figure 9). This finding suggests that anti-HMGB1 mAb administration effectively inhibits I/R-induced detachment of astrocyte endfeet from the vascular wall.

### 2.7. Determination of Inflammation-Related Molecules in the Spinal Cord Under I/R Injury

To investigate the potential anti-inflammatory mechanisms of anti-HMGB1 mAb treatment, we analyzed the expression of inflammation-related genes in the gray matter of the spinal cord 48 h after I/R injury (Figure 10). When all data were included (*n* = 5 per group), no statistically significant differences were detected between the control IgG and anti-HMGB1 mAb groups for HMGB1 (*p* = 0.22), NF-κB (*p* = 1.00), IL-1β (*p* = 0.095), IL-6 (*p* = 0.69), TNF-α (*p* = 0.31), or TLR4 (*p* = 0.15). Effect size estimates indicated small-to-moderate negative Cliff’s δ values for HMGB1 (−0.52), IL-1β (−0.68), TNF-α (−0.44), and TLR4 (−0.60), suggesting a trend toward reduced expression in the anti-HMGB1 mAb group, although confidence intervals were wide (Appendix A). Sensitivity analyses excluding IQR-flagged outliers showed similar trends but likewise failed to reach statistical significance (Appendix A).

## 3. Discussion

SCI/R is characterized by a biphasic pathophysiological process consisting of an initial ischemic insult followed by secondary injury mechanisms upon reperfusion. The primary ischemic phase induces energy depletion, ionic imbalance, and excitotoxicity, resulting in neuronal and glial injury. The reperfusion phase exacerbates tissue damage through oxidative stress, mitochondrial dysfunction, inflammatory cell infiltration, and disruption of the BSCB [23]. Among the key mediators of these secondary processes, HMGB1 acts as a prototypical damage-associated molecular pattern (DAMP), linking ischemic cell death to sustained neuroinflammation [15,24,25]. Accumulating evidence indicates that HMGB1 engages multiple intracellular pathways across neurons, glial cells, and infiltrating immune cells during SCI/R. In neurons, early oxidative stress and calcium overload rapidly activate stress-responsive kinases such as p38 MAPK, JNK, and NF-κB, leading to post-translational modifications of HMGB1 (acetylation and phosphorylation). Such modifications disrupt nuclear retention signals, driving its translocation to the cytoplasm and subsequent extracellular release [26,27,28]. The release of HMGB1 follows a cell death–dependent pattern, occurring passively during necrosis and in a more restricted manner during early apoptosis [29]. Upon stimulation with proinflammatory cytokines (e.g., interleukin (IL)-1β, tumor necrosis factor (TNF)-α), astrocytes activate the TLR4–MyD88–NF-κB and STAT3 pathways, driving active HMGB1 secretion [30,31]. Astrocyte-derived HMGB1 exerts dual effects: while it promotes neurovascular remodeling and repair, excessive release increases BSCB permeability and facilitates leukocyte infiltration [30,32]. Microglia are activated within hours of I/R injury, responding to HMGB1 via TLR4 and RAGE. This activation further amplifies inflammatory cascades through p38 MAPK and JAK/STAT signaling, leading to the release of secondary cytokines such as IL-1β, IL-6, and TNF-α [33,34]. Infiltrating immune cells, including neutrophils and macrophages, also release HMGB1 through mechanisms such as NLRP3 inflammasome activation and oxidative DNA damage [35,36]. Neutrophils constitute the predominant source in the acute phase, whereas macrophages sustain HMGB1 release during the subacute and chronic phases [37]. Collectively, these findings highlight the cell type–specific signaling pathways through which HMGB1 orchestrates inflammatory amplification and tissue degeneration following SCI/R. In contrast to spinal cord ischemia–reperfusion injury (SCI/R), traumatic spinal cord injury (TSCI) is initiated by mechanical disruption of neural and vascular structures, leading to immediate axonal damage, hemorrhage, and rupture of the blood–spinal cord barrier (BSCB) [38,39]. SCI/R follows a biphasic course with a delayed component driven by reperfusion injury [40], whereas TSCI presents with immediate neurological deficits caused by irreversible structural damage, followed by secondary changes such as edema, ischemia, and inflammatory cell infiltration [38,39,41]. These differences highlight the need for distinct therapeutic approaches: hemodynamic optimization and anti-inflammatory strategies in SCI/R, versus early decompression and stabilization in TSCI [38,39]. In terms of frequency, postoperative paraplegia after TAAA repair occurs in approximately 3–4% of patients [3], whereas traumatic spinal cord injuries due to contusion or section have an annual incidence of 10–50 per million population, with paraplegia observed in about 30–40% of cases [42,43].

In the present study, we used a rabbit model of SCI/R injury induced by abdominal aortic clamping to mimic postoperative delayed paraplegia after TAAA surgery. Our results demonstrated that administration of an anti-HMGB1 mAb significantly improved neurological outcomes and preserved motor neuron survival in the anterior horn. These effects were associated with the suppression of microglial activation, oxidative stress, BSCB hyperpermeability, and neuronal apoptosis. Similar neuroprotective effects of anti-HMGB1 mAbs have been reported in rodent models of ischemic brain injury, intracranial hemorrhage, and traumatic brain injury [15,16,17,18], as well as in a mouse model of spinal cord injury [44]. Taken together, these findings support the potential of anti-HMGB1 mAb treatment as a novel therapeutic strategy for preventing postoperative paraplegia following TAAA surgery.

The therapeutic effects of anti-HMGB1 mAbs are hypothesized to be primarily mediated through the reduction in HMGB1 concentrations in both the circulation and spinal cord tissue. Although the number of HMGB1-positive cells was significantly lower in the spinal cord of the anti-HMGB1 mAb group than in the control IgG group, the Plasma HMGB1 levels were not significantly different between the two groups. This discrepancy may be attributed to the highly localized nature of ischemic injury in this model, resulting in a relatively small volume of spinal cord tissue subjected to I/R damage. Moreover, within the lumbar segments caudal to L5, the actual extent of tissue exposed to ischemia–reperfusion varied among individual animals, reflecting differences in collateral circulation. Such variability in the distribution and amount of affected tissue likely limited the systemic release of HMGB1, thereby contributing to the absence of a significant elevation in plasma levels. HMGB1 exacerbates I/R injury through multiple mechanisms, particularly by enhancing inflammatory signaling through the TLR4 [19] and receptor for advanced glycation end-products (RAGE) pathways [45]. HMGB1 also modulates intracellular signaling cascades such as the phosphoinositide 3 kinase (PI3K)/Akt [46] and mitogen-activated protein kinase (MAPK) pathways [47], influencing cellular survival, proliferation, and differentiation, and ultimately contributing to the pathogenesis of I/R injury. Previous studies have reported that pharmacological agents such as dexmedetomidine [19], ethyl pyruvate [20], and glycyrrhizin [21,22] act as HMGB1 inhibitors and effectively attenuate SCI/R injury. However, to our knowledge, the present study is the first to directly target HMGB1 using a mAb and to evaluate its therapeutic potential in this context. Although further studies are required to determine the optimal HMGB1-targeting strategy with the most favorable risk-benefit profile for clinical applications, antibody-based inhibition may offer superior specificity and targeted therapeutic action.

A 48 h observation period was selected for this study on the basis of previous reports indicating that this timeframe was sufficient to detect the onset of delayed spinal cord injury in rabbits. Matsumoto et al. reported the development of delayed motor dysfunction in most rabbits at 48 h following SCI, which was associated with poor recovery of segmental spinal cord evoked potentials as early as 15 min after reperfusion [48]. Similarly, Moore et al. observed delayed spinal cord injury in all rabbits within 14 to 48 h post-I/R insult [49]. Kawanishi et al. also documented the presence of spinal cord injury in all animals 48 h after I/R [50]. Consequently, the 48 h observation period has been widely adopted as a standard to assess delayed neurological damage in rabbit models of SCI/R injury [51,52].

In the present study, motor neuron survival in the anterior horn significantly improved in the anti-HMGB1 mAb group. However, the analysis was complex because of anatomical considerations. Aortic occlusion was performed just distal to the renal arteries, typically below the left renal artery, which predominantly induced ischemia in the spinal cord below the L5 level. On the basis of this factor, we initially quantified motor neuron survival at the L7 level, where ischemic injury was expected to be the most pronounced. Although a trend toward improvement was observed, statistical significance was not reached. Nonetheless, a Cliff’s delta value of +0.84 indicated a strong effect size, prompting further analysis at the L5 and L6 levels. Statistical significance was achieved when data from all three levels were included. Interestingly, the variability in motor neuron survival decreased at more distal spinal levels. This may reflect individual differences in the collateral blood supply originating above L4. In animals with well-developed collaterals, proximal spinal segments were more likely to be preserved. However, rabbits have less extensive spinal collaterals than humans, who rarely develop SCI after abdominal aortic clamping. Therefore, in rabbits, perfusion to distal segments may remain insufficient even in well-perfused animals, resulting in more consistent injury in those regions. Although the significance was more difficult to demonstrate in the results for neuronal counts, paraplegic symptoms were consistent across animals, supporting the validity of the outcomes. Furthermore, the expression of cleaved caspase-3, a marker of cell death, was significantly lower in the anti-HMGB1 mAb group than in the control IgG group. These findings indirectly support the neuroprotective effect of the anti-HMGB1 mAb in suppressing motor neuron apoptosis. To address the anatomical variability, we performed level-specific histological quantification and section-based statistical analyses. While increasing the sample size could improve the statistical power, we deemed the current results sufficiently robust on the basis of symptom consistency and the findings of sub-analyses. From an animal welfare perspective, further animal use was considered unnecessary. Notably, individual variations in spinal collateral circulation should be carefully considered when evaluating SCI/R injury models.

Our findings demonstrated that administration of an anti-HMGB1 mAb significantly attenuated microglial activation, neutrophil infiltration, and the expression of 4-HNE, a lipid peroxidation product and an established biomarker of oxidative stress [53,54]. Oxidative stress and inflammation are closely interconnected processes that play pivotal roles in the pathogenesis of I/R injury. Oxidative stress arises from the excessive accumulation of ROS, which, although essential for cellular signaling, can interact with critical biological macromolecules, such as lipids, proteins, and DNA, to disrupt cellular homeostasis and trigger cell death [55]. In addition, I/R injury markedly elevated the expression of proinflammatory cytokines in the ischemic spinal cord, initiating a cascade of harmful events including vasomotor dysregulation, microvascular obstruction, cytotoxic enzyme release, and further ROS production [56].

In this study, the expression of inflammation-related genes was quantified using real-time PCR. When all data were included in the analysis, no statistically significant differences were detected between the control IgG and anti-HMGB1 mAb groups for HMGB1, TLR4, NF-κB, IL-1β, IL-6, or TNF-α. Nevertheless, HMGB1 and TLR4 showed a tendency toward lower expression in the anti-HMGB1 mAb group, consistent with the proposed mechanism of HMGB1 blockade. The absence of statistical significance may be partly attributable to the inter-individual variability in collateral blood flow, as previously discussed, and to the limited sample size. In addition, differences in the temporal expression profiles of these cytokines may have contributed, since the 48 h time point examined in this study may not coincide with the peak expression of some inflammatory mediators. Furthermore, the qPCR analysis was performed on homogenized gray matter obtained from the residual tissue after histological and Western blot assessments, consisting mainly of the L5 and L7 segments with smaller amounts from L6. The relative contribution of each segment was not constant across animals, which may have introduced variability and masked localized differences in gene expression. This pooling approach may therefore have underestimated region-specific inflammatory responses. Sensitivity analyses excluding IQR-defined outliers yielded similar trends but likewise did not reach statistical significance, underscoring the exploratory nature of these findings. To aid interpretation, effect size estimates are provided in Appendix A and confidence intervals in Appendix A.

Regarding microglial responses, our immunofluorescence analysis revealed that only a few microglial cells exhibited double labeling of Iba1 and HMGB1 at 48 h after I/R injury. Several factors may account for this observation. First, HMGB1 is a nuclear protein that is rapidly translocated and released extracellularly from neurons and astrocytes under stress or injury, whereas such pronounced nuclear-to-cytoplasmic translocation does not always occur in microglia. Second, microglia generally express relatively low basal levels of HMGB1, and they are often considered to function primarily as responders through HMGB1 receptor signaling rather than as major sources of HMGB1. Third, HMGB1 expression dynamics are time dependent; robust microglial HMGB1 expression may appear during subacute or chronic phases, whereas our analysis was limited to the acute phase within 48 h. Finally, methodological aspects should be considered, as Iba1 is a cytoplasmic marker whereas HMGB1 localizes to both the nucleus and cytoplasm, making colocalization more difficult to detect. Taken together, these factors may explain why only a small number of Iba1/HMGB1 double-positive cells were observed in this study.

In this study, we demonstrated that I/R injury induces structural damage throughout the anterior horn of the spinal cord, including a reduction in viable motor neurons and activation of microglia, along with disruption of the BSCB. We also observed microvascular leakage of albumin, a well-established indicator of BSCB disruption, which was significantly attenuated by treatment with the anti-HMGB1 mAb. Notably, we had previously reported a similar protective effect of anti-HMGB1 mAbs on the BBB in the brain, and the present findings confirm that analogous mechanisms are involved in the spinal cord as well [57]. While previous studies have suggested that HMGB1-mediated spinal cord injury is primarily driven by inflammation through activated microglia and macrophages, rather than by astrocyte dysfunction [58], our present findings indicated a reduction in perivascular astrocytic processes and their detachment from microvessels following I/R injury, resulting in the formation of a gap between the astrocyte endfeet and the vessel wall. This pathological alteration was markedly reduced by the anti-HMGB1 mAb treatment, indicating that the antibody may improve astrocytic function in the injured spinal cord. The BBB, which shares a similar structural organization with the BSCB, is composed of vascular endothelial cells, pericytes, and astrocytic endfeet, and plays a vital role in maintaining CNS homeostasis [59,60]. BSCB disruption may facilitate the infiltration of neurotoxic plasma components and immune cells, thereby exacerbating neurological deficits. Previous studies have demonstrated that BBB breakdown plays a central role in various CNS disorders, including stroke, traumatic brain injury, and intracerebral hemorrhage [61,62]. In support of these findings, Zhang et al. used electron microscopy to demonstrate that astrocyte endfeet underwent swelling and detachment from the basal lamina following cerebral I/R injury, identifying these features as hallmarks of BBB disruption [18]. Using an in vitro BBB model, the same group further showed that administration of recombinant HMGB1 increased BBB permeability to Evans blue-albumin by inducing morphological and functional changes in endothelial cells and pericytes. Consistent with these findings, our previous study demonstrated that treatment with an anti-HMGB1 mAb significantly ameliorated brain infarction in a rat model of middle cerebral artery occlusion by preserving BBB integrity through suppression of microglial activation, TNF-α and inducible nitric oxide synthase expression, and matrix metalloproteinase-9 activity, without altering cerebral blood flow [15]. Together with the present results, these findings suggest that the anti-HMGB1 mAb may improve neurological outcomes following SCI/R injury by preserving BSCB integrity through mechanisms analogous to those observed in BBB-related CNS pathologies.

Recent studies have employed abdominal aortic occlusion times of 15 to 30 min to establish rabbit models of SCI/R injury [19,20,63]. However, we could not replicate these models under our experimental conditions. Our results showed that clamping the abdominal aorta for 12–30 min consistently resulted in complete paraplegia within 6 h after reperfusion, despite maintaining stable intraoperative circulatory parameters. Notably, even treatment with anti-HMGB1 mAb failed to reverse the paralysis under these conditions. Through iterative adjustments of the ischemic duration, we found that an 11 min occlusion period was optimal for inducing delayed paraplegia in rabbits. Although this ischemic duration is relatively shorter than those reported in previous models, variations in factors such as animal source, body temperature, body weight, and occlusion technique may have influenced ischemic tolerance. Our 11 min ischemia model represents a reproducible and physiologically relevant platform for studying delayed paraplegia following SCI/R injury and offers a suitable preclinical framework for assessing the efficacy of anti-HMGB1 mAb administration as a therapeutic intervention.

In this study, we demonstrated the delayed neuroprotective effects of an anti-HMGB1 mAb in a rabbit model of SCI/R injury. Further investigations are warranted to optimize the therapeutic applications of this approach. Although the anti-HMGB1 mAb was administered after the onset of I/R in our protocol, determining whether prophylactic administration before ischemia could more effectively prevent spinal cord injury and reperfusion-related damage is a topic of interest. Spinal damage is commonly observed in clinical settings such as thoracic aortic surgery, raising the possibility that intrathecal administration of anti-HMGB1 mAb may be a feasible and potentially superior route for spinal cord protection. Should intrathecal delivery prove to be more effective than intravenous administration, this finding could provide important implications for clinical practice. In addition, HMGB1 blockade has been reported to attenuate neuropathic pain in various preclinical models, suggesting that anti-HMGB1 mAb may also be beneficial in managing pain associated with spinal cord injury [64,65,66]. Continued research is essential to further elucidate the therapeutic potential of anti-HMGB1 mAb administration and facilitate its translation into clinical applications. To date, however, no anti-HMGB1 mAb has entered clinical trials for CNS or other indications, and potential concerns such as immunogenicity, infection risk, interference with reparative HMGB1 functions, and long-term risk-benefit must be carefully addressed, underscoring both the promise and the current translational gap of this strategy. This study establishes a robust and clinically relevant large-animal model that can serve as a platform for subsequent efficacy and safety testing, and provides exploratory proof-of-concept evidence supporting anti-HMGB1 antibody therapy as a potential preventive or therapeutic strategy for delayed spinal cord injury after aortic surgery, thereby underscoring the preclinical and clinical relevance of our findings in bridging experimental research to future clinical translation.

This study had several limitations beyond those previously discussed. First, although circulatory and respiratory parameters were stabilized as much as possible during surgery, as shown in Table 2, they were not completely uniform across all animals. Second, the postoperative observation period was limited to 48 h. As a result, the results did not account for delayed spinal cord injury or improvements in the existing deficits that appear after 48 h. The relevance of the 48 h endpoint was addressed earlier. Third, individual differences in collateral circulation among rabbits, which may have influenced the extent of spinal cord I/R injury, were not evaluated in this study. Fourth, only male rabbits were used in the experiments. Potential sex-related differences in vascular physiology, inflammatory responses, and susceptibility to ischemia–reperfusion injury were therefore not addressed, and future studies including both sexes will be necessary to assess the generalizability of the present findings.

## 4. Materials and Methods

### 4.1. Rabbit Model of SCI/R Injury Using Aortic Cross Clamp

The animal study protocol was approved by the Institutional Animal Care and Use Committee of Okayama University (protocol code: OKU-2020410; date of approval: 25 June 2020). All animals were cared for in strict accordance with the recommendations in the Guidelines for the Care and Use of Laboratory Animals issued by the National Institutes of Health, ARRIVE guidelines, and the regulations for animals in Japan (Standards relating to the Care and Keeping and Reducing Pain of Laboratory Animals: Notice of the Ministry of the Environment No. 88 of 2006 and the latest revision: Notice of the Ministry of the Environment No. 84 of 2013) [67].

Male New Zealand white rabbits (Japan SLC Inc., Shizuoka, Japan) weighing 2.0–2.2 kg were used in the experiments. The rabbits were anesthetized using midazolam (0.2 mg/kg) (Midazolam Sandoz, Sandoz K.K., Tokyo, Japan) intramuscular injection and isoflurane inhalation (4% for induction, 2–3% for maintenance) (Isoflurane, Viatris Inc., Tokyo, Japan). Spontaneous breathing was maintained, and oxygen was administered through an anesthetic face mask. An intravenous line was established in the ear vein for continuous infusion of lactated Ringer’s solution (5 mL/kg/h). Blood oxygen saturation (SpO_2_) was percutaneously monitored in the ear opposite the location where the venous line was secured. A 24-gauge catheter (Terumo, Tokyo, Japan) was inserted into the left femoral artery for distal blood pressure monitoring. The rectal temperature was maintained at 38.5 °C ± 0.5 °C using a heating blanket (Warm Reversible Heater, MARUKAN Co., Ltd., Osaka, Japan) throughout the operation. The rabbits were positioned supine, and an approximately 5 cm-long abdominal median incision was made to expose the abdominal aorta at the level of the left renal artery (a reduction in spinal cord arterial blood flow is commonly observed at or below the L5 level). The aorta was occluded with micro-bulldog clamps for 11 min after administration of heparin (100 mg/kg) (Heparin Sodium Injection, Mochida Pharmaceutical Co., Ltd., Tokyo, Japan). After removing the clamps, the abdominal wall was closed, and the animals were returned to their cages.

All rabbits were housed in individual enclosures measuring at least 0.3844 square meters per rabbit. The environment was maintained with a 12 h light/dark cycle with comfortable seasonal temperature (16 °C to 25 °C) and humidity (approximately 50%). These conditions were carefully controlled to minimize stress and promote the health of the rabbits. The animals were provided with food and water ad libitum. The cleanliness of the cages was monitored daily and maintained by the caretakers.

### 4.2. Experimental Protocols

The anti-HMGB1 mAb and the class-matched control IgG used in this study were generated as previously described by Liu et al. [15].

The rabbits were administered either anti-HMGB1 mAb (#10-22; IgG2a subclass; 2 mg/kg intravenously) or class-matched control IgG (anti-keyhole limpet hemocyanin mAb) twice, immediately and 6 h after reperfusion. Motor function in the hind limbs was assessed at 6, 24, and 48 h after reperfusion using the modified Tarlov score [19]. The scoring criteria were as follows: 0, no voluntary hind limb function; 1, only perceptible joint movement; 2, active movement but unable to stand; 3, able to stand but unable to walk; and 4, complete normal hind limb motor function. At 48 h after reperfusion, the rabbits were euthanized using 150 mg/kg pentobarbital administration intravenously, and spinal cord and blood samples were obtained. The lumbar spinal cord segments (L5–L7) were excised. For spinal cord harvesting, the abdominal cavity was opened to approach the anterior aspect of the spine. The L7 vertebra, located directly above the pelvis, was identified, and three consecutive vertebrae together with the enclosed spinal cord were removed. The extracted vertebrae were subsequently dissected from the posterior side to access the spinal canal, and the spinal cord was exposed. The cord was then transected between vertebrae, and segments corresponding to the L5, L6, and L7 levels were carefully harvested. Tissue samples from the L5, L6, and L7 spinal cord segments were partially preserved in 10% formalin for histological evaluation, and the remaining portions were homogenized for Western blot and PCR analyses.

### 4.3. Hematoxylin and Eosin Staining

Formalin-fixed L5, L6, and L7 segment spinal cord tissues were thoroughly washed in PBS for 24 h with several buffer changes to remove residual fixative. Samples were then dehydrated through a graded ethanol series (50%, 70%, 80%, 90%, 95%, and 100% × 3; ~4 h each), cleared in toluene (two changes, ~4 h each), and infiltrated with paraffin wax at 60 °C. Paraffin infiltration was performed stepwise, beginning with a toluene–paraffin mixture (1:1, 2 h), followed by two changes in pure paraffin (4 h each). Finally, tissues were embedded in pre-warmed paraffin molds and rapidly cooled at room temperature. Axial sections were cut at 5 μm thickness using a rotary microtome (HM325, Thermo Fisher Scientific, Waltham, MA, USA).

Hematoxylin–eosin staining was performed at the Central Research Laboratory, Okayama University, using their standard protocol. Briefly, paraffin sections were deparaffinized in xylene, rehydrated through a graded ethanol series, stained with hematoxylin, rinsed and blued in running tap water, counterstained with eosin, dehydrated in graded ethanol, cleared in xylene, and mounted with a coverslip.

Spinal cord morphology was examined using LSM 780 confocal microscope (Carl Zeiss Inc., Thornwood, NY, USA), and injured neurons were identified by the presence of eosinophilic cytoplasm, pyknotic nuclei, and the reduced presence or absence of Nissl bodies. Normal neurons exhibited a polygonal shape with prominent Nissl bodies. Pathological changes in Nissl bodies were defined as loss of clear boundaries, decreased staining intensity, and reduction in size compared with those of normal neurons. One transverse section per spinal cord level (L5, L6, and L7) was prepared, and all viable motor neurons located within the bilateral anterior horns of each section were counted and included in the analysis.

### 4.4. Immunofluorescence Staining

Paraffin-embedded L6 spinal cord sections were prepared for immunofluorescence staining as described in our previous study [15]. Five rabbits from each group were treated with control IgG or the anti-HMGB1 mAb. The primary antibodies used in the experiments were mouse anti-HMGB1 mAb (1:200; R&D Systems Inc., Minneapolis, MN, USA), rabbit anti-Iba1 polyclonal antibody (pAb) (1 μg/mL; Wako, Osaka, Japan), rabbit anti-MPO pAb (1:50; Abcam, Cambridge, UK), sheep anti-albumin pAb (1:200; Bethyl Laboratory, Montgomery, TX, USA), rabbit anti-GFAP pAb (1:100; Abcam, Cambridge, UK), and rabbit anti-PDGFRβ mAb (1:100; Abcam, Cambridge, UK). The sections were counterstained with 4′,6-diamidino-2-phenylindole (DAPI) (300 μM, 1:1000; Thermo Fisher Scientific Inc., Waltham, MA, USA) and incubated with secondary antibodies conjugated to Alexa-488, Alexa-555, or Alexa-594 (1:500; Thermo Fisher Scientific Inc., Waltham, MA, USA). The sections were then mounted using the VECTASHIELD Hard Set Mounting Medium with DAPI (Vector Laboratories Inc., Newark, CA, USA) and observed under an LSM 780 confocal microscope (Carl Zeiss Inc., Thornwood, NY, USA). Microglia and MPO-positive neutrophils were counted, and the area of HMGB1-positive cells was quantified using ImageJ 1.42q software (NIH, Bethesda, MD, USA).

### 4.5. Western Blot Analysis

Spinal cord samples from the L6 segment were sliced to a thickness of 3 mm, homogenized for 5 min in radio-immunoprecipitation assay lysis buffer containing a cocktail of protease inhibitors (MilliporeSigma, Burlington, MA, USA), and separated by 12% sodium dodecyl-sulfate polyacrylamide gel electrophoresis. The proteins were then transferred onto nitrocellulose membranes. After blocking with 10% skim milk, the membranes were incubated at 4 °C overnight with horseradish peroxidase-labeled rat anti-HMGB1 mAb (prepared in our laboratory), rabbit anti-caspase-3 pAb (1:1000; Cell Signaling Technology Inc., Danvers, MA, USA), and rabbit anti-4-HNE pAb (1:1000; Abcam, Cambridge, UK). β-Actin, used as a reference protein, was probed with a mouse anti-β-actin mAb (1:1000; Santa Cruz Biotechnology, Santa Cruz, CA, USA), followed by incubation with a goat anti-mouse antibody (1:5000). Finally, the bands were visualized using an ECL system (Thermo Fisher Scientific Inc., Waltham, MA, USA) and analyzed using ImageJ software (version 1.53t).

### 4.6. Enzyme-Linked Immunosorbent Assay

Plasma HMGB1 levels were determined using blood samples (1 mL) collected from the hearts of deeply anesthetized rabbits into ethylenediaminetetraacetic acid tubes and centrifuged for 10 min at 1500× *g*. The supernatants, supplemented with protease inhibitor (Sigma-Aldrich, St. Louis, MO, USA) (50 μL/g), were stored at −20 °C until analysis. HMGB1 was detected using an ELISA kit (Shino-Test Co., Tokyo, Japan) according to the manufacturer’s instructions.

### 4.7. Quantitative Real-Time PCR

Real-time PCR was performed as described previously [15] using SYBR Premix EX Taq (Takara Bio, Shiga, Japan) on a LightCycler instrument (Roche, Basel, Switzerland), according to the manufacturer’s instructions. Analyses were conducted using the spinal cord tissues remaining after histological examination (spinal cord tissue from L5 and L7 segments were mainly used), ELISA, and Western blot had been completed. The tissues were carefully dissected to remove as much white matter as possible, and the remaining gray matter was homogenized for RNA extraction and subsequent analysis. The sense and antisense primers used to analyze messenger RNA (mRNA) expression are listed in Appendix A. GAPDH was used as an internal control to normalize cDNA levels. Fold changes in expression levels were calculated using the comparative cycle threshold method (2^–ΔΔCt^).

### 4.8. Statistical Analysis

Continuous variables were represented as the mean ± standard error of the mean (SEM). Intraoperative blood pressure, heart rate, SpO_2_, body temperature values, pre- and post-operative modified Tarlov score, and the results of quantitative real-time PCR were compared between the control IgG group and the anti-HMGB1 mAb group using the exact Mann–Whitney U test. An unpaired *t*-test was used to analyze plasma HMGB1 concentration, the number of viable motor neurons, the number of microglia per animal, the number of MPO-positive neutrophils, and the relative intensities of cleaved caspase-3 and 4-HNE. The significance level was set at *p* < 0.05. To complement hypothesis testing, effect sizes were estimated using Cliff’s delta with 95% bootstrap confidence intervals, and Hodges–Lehmann median differences were also calculated. Outliers in the PCR dataset were identified a priori using the interquartile range (IQR) method (values < Q1 − 1.5 × IQR or >Q3 + 1.5 × IQR), but all values were retained in the primary analysis; sensitivity analyses excluding IQR-flagged values are presented in the Appendix A.

Although a Sham group was included for reference, statistical comparisons with the Sham group were not performed because the sample size was only 1–2 animals and any differences from the control or anti-HMGB1 mAb groups were visually evident; thus, such comparisons were considered of limited statistical value. In principle, two animals were assigned to the Sham group; however, in some experiments (Figure 4 and Figure 7A,B), only one sample was available due to limited tissue availability during harvesting.

## 5. Conclusions

We established a rabbit model of delayed SCI/R injury using an optimized 11 min occlusion protocol, which faithfully replicated the delayed-onset paraplegia observed after TAAA surgery. Using this model, we demonstrated that administration of an anti-HMGB1 mAb significantly ameliorated delayed SCI/R injury. The protective effects appeared to be mediated through the neuroprotective, anti-inflammatory, and antioxidant properties of the antibody, as well as its capacity to preserve BSCB integrity. Importantly, this exploratory proof-of-concept study provides preclinical evidence supporting anti-HMGB1 mAb administration as a potential preventive or therapeutic strategy for delayed spinal cord injury after aortic surgery, thereby underscoring its translational relevance as a bridge toward future efficacy and safety studies in humans.

## Figures and Tables

**Figure 1 ijms-26-08643-f001:**
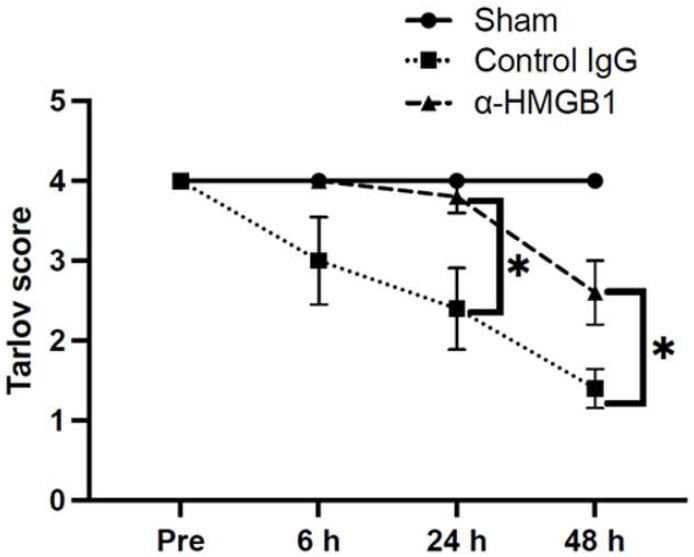
Neurological outcomes. Neurological outcomes were measured at the indicated times in each group using a modified Tarlov score. In the control IgG group, clear neurological deficits developed after I/R injury and showed a progressive worsening at 24 and 48 h. Treatment with α-HMGB1 significantly ameliorated these deficits compared with the control IgG group. Furthermore, no neurological deficits were observed in the Sham group. Values are shown as means ± SEM in the control IgG group (*n* = 5) and α-HMGB1 group (*n* = 5). The Sham group (*n* = 2) is shown as a reference only, with the median reported for two subjects; no statistical comparisons were performed due to the limited sample size. α-HMGB1, anti-HMGB1 monoclonal antibody; HMGB1, high mobility group box 1. * Statistical significance: * *p* < 0.05.

**Figure 2 ijms-26-08643-f002:**
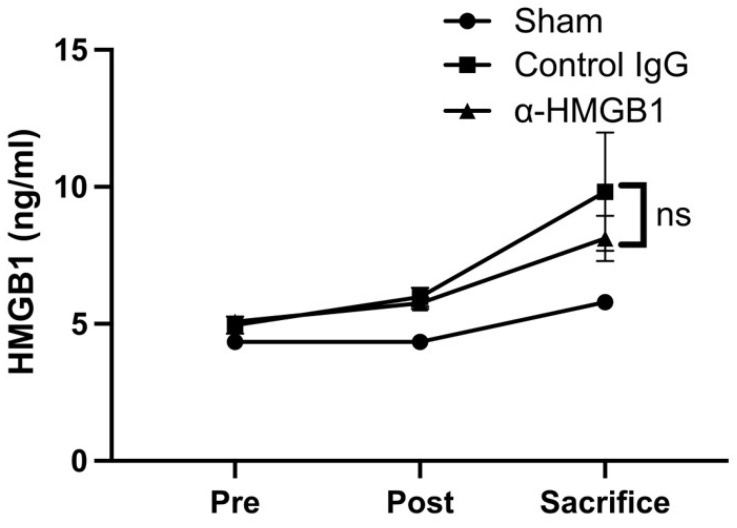
Plasma levels of HMGB1. Plasma levels at the indicated times were determined using ELISA in the Sham group (*n* = 2), the control IgG group (*n* = 5), and the α-HMGB1 group (*n* = 5). HMGB1 levels progressively increased after surgery in both experimental groups, with no significant difference between the α-HMGB1 mAb and control IgG groups. In the Sham group, a slight increase was observed but values remained nearly equivalent to preoperative levels. The Sham group is shown as a reference only, and the median is reported for two subjects; no statistical comparisons were performed due to the limited sample size. Values represent means ± SEM. α-HMGB1, anti-HMGB1 monoclonal antibody; HMGB1, high mobility group box 1; ns, not significant.

**Figure 3 ijms-26-08643-f003:**
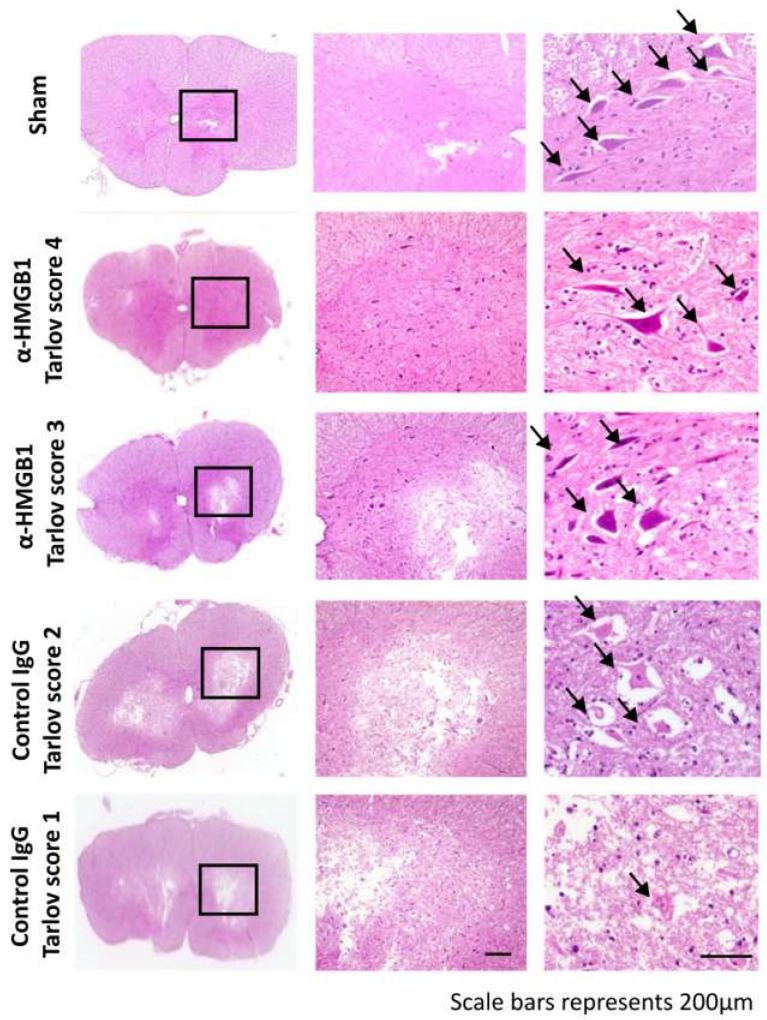
Hematoxylin–eosin staining of rabbit spinal cord tissue of each Tarlov score after I/R injury and that of the Sham group. The pictures in the left column show whole images of the spinal cord tissues. The boxed areas in the left column are magnified in the middle column, which depicts the anterior horn region with various infarction areas corresponding to each Tarlov score. The right column presents representative high-magnification images of motor neurons. Normal motor neurons exhibit large, darkly stained, polygonal structures with well-defined Nissl bodies (black arrows). As the Tarlov score decreases, pathological changes become evident, including blurred boundaries, paler staining, and reduction in the size of Nissl bodies. α-HMGB1, anti-HMGB1 monoclonal antibody; HMGB1, high mobility group box 1; I/R, ischemia–reperfusion.

**Figure 4 ijms-26-08643-f004:**
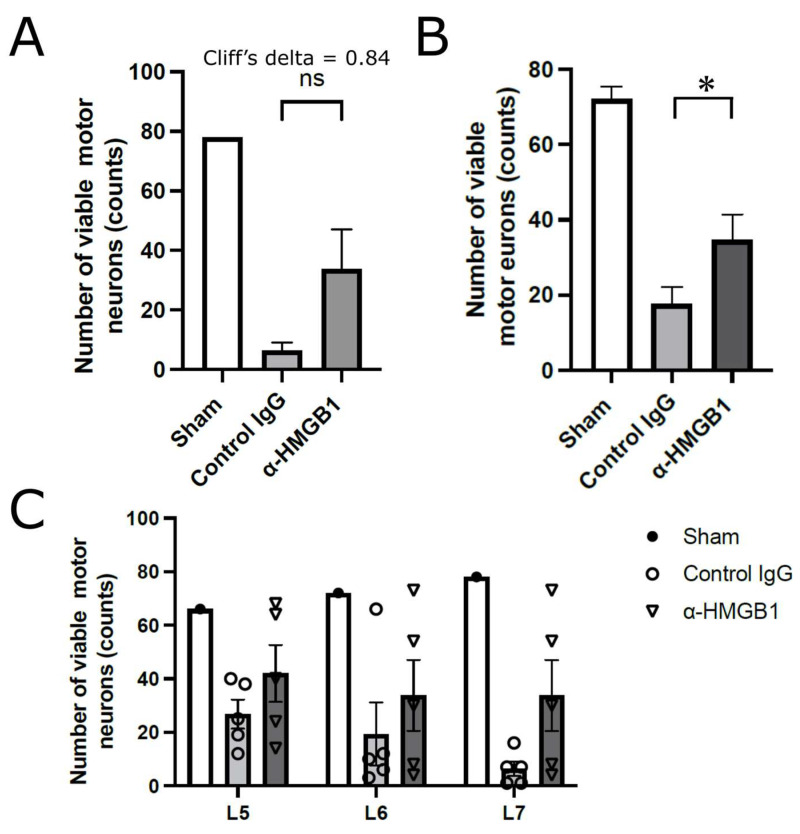
Viable neuron analysis in the anterior horns at three spinal cord levels per animal, 48 h after I/R injury. The number of evaluated animals was one in Sham group, five in control IgG group, and five in α-HMGB1 group. Viable motor neurons were counted in the anterior horns at the L5, L6, and L7 levels, using one transverse section per animal. The Sham group is shown as a reference only; no statistical comparisons were performed due to the limited sample size. (**A**) Comparison of viable motor neuron counts at the L7 level. Although not statistically significant (*p* = 0.126), the Cliff’s delta value (+0.84) indicated a very strong effect size. (**B**) Comparison of total viable anterior horn motor neurons per animal, calculated from sections at L5–L7. The α-HMGB1 group showed significantly higher counts than the control IgG group (* *p* < 0.05), indicating a neuroprotective effect. (**C**) Individual animal data for each spinal level (L5, L6, L7). In the control IgG and α-HMGB1 groups (*n* = 5 each), no statistically significant differences were detected at individual levels. Some animals exhibited unusually high numbers of surviving neurons, likely reflecting inter-individual variability in collateral circulation. Such outliers were less frequent at L7 compared with L5 and L6, consistent with reduced collateral supply at more distal levels. α-HMGB1, anti-HMGB1 monoclonal antibody; HMGB1, high mobility group box 1; I/R, ischemia–reperfusion; ns, not significant.

**Figure 5 ijms-26-08643-f005:**
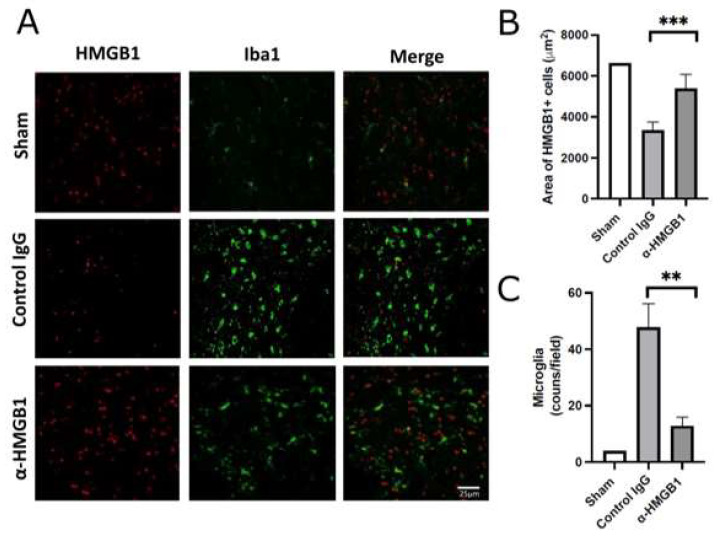
Immunofluorescence analysis of HMGB1-positive cells and microglia (Iba1-positive cells) in the anterior horn of the spinal cord. (**A**) Spinal cords were harvested 48 h after I/R injury and double-stained with α-HMGB1 (red) and anti-Iba1 (green). Iba1-positive cells in the control IgG group showed short processes and enlarged cell bodies. Representative anterior horn images are shown. Scale bar = 25 μm. (**B**) Quantification of HMGB1-positive area in the anterior horn at 48 h after reperfusion. Data were obtained from the central region of ×100 magnified fields in the Sham group (*n* = 2), the control IgG group (*n* = 5), and the α-HMGB1 group (*n* = 5). The Sham group is shown as a reference only; no statistical comparisons were performed due to the limited sample size. Values represent means ± SEM. (**C**) Quantification of Iba1-positive microglial cells in the anterior horn across the same groups. Values represent means ± SEM. α-HMGB1, anti-HMGB1 monoclonal antibody; HMGB1, high mobility group box 1; Iba1, ionized calcium-binding adaptor molecule 1. * Statistical significance: ** *p* < 0.01, *** *p* < 0.001.

**Figure 6 ijms-26-08643-f006:**
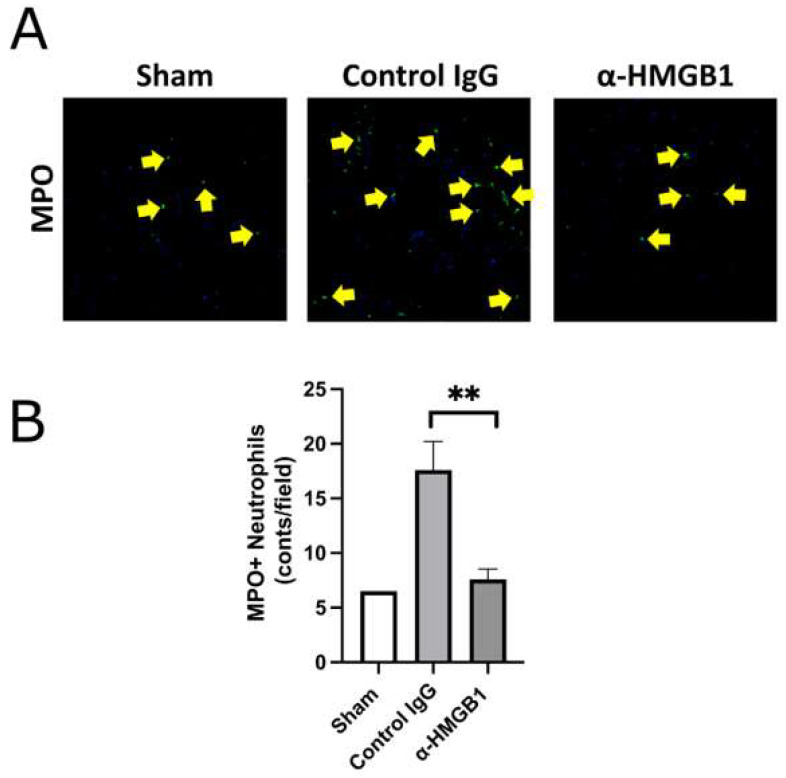
Quantification of MPO-positive area in the anterior horn of the spinal cord 48 h after reperfusion. (**A**) Representative spinal cord sections stained with MPO. Scale bar = 50 μm. (**B**) Quantification of MPO-positive neutrophils (arrows) in the anterior horn, obtained from the central region at ×200 magnification in the Sham group (*n* = 2), control IgG group (*n* = 5), and α-HMGB1 group (*n* = 5). The Sham group is shown as a reference only; no statistical comparisons were performed due to the limited sample size. MPO-positive neutrophils were markedly increased in the control IgG group compared with the Sham group, whereas this increase was significantly suppressed by anti-HMGB1 monoclonal antibody administration, to levels comparable to the Sham group. Values represent means ± SEM. α-HMGB1, anti-HMGB1 monoclonal antibody; HMGB1, high mobility group box 1; MPO, myeloperoxidase. ** *p* < 0.01.

**Figure 7 ijms-26-08643-f007:**
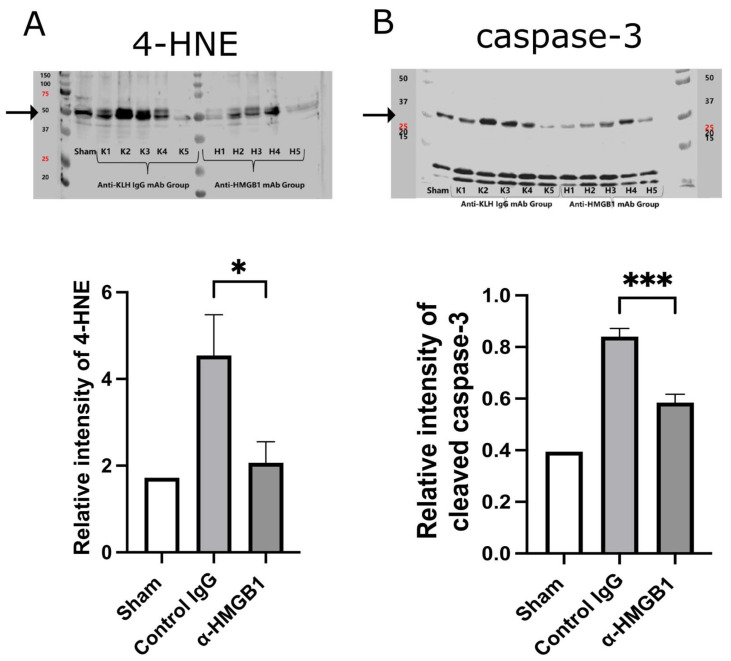
Effects of anti-HMGB1 monoclonal antibody on oxidative stress and apoptosis in the spinal cord at 48 h after I/R injury. Effects of anti-HMGB1 monoclonal antibody (α-HMGB1) on oxidative stress and apoptosis in the spinal cord at 48 h after I/R injury. (**A**) Relative intensity of 4-hydroxynonenal (4-HNE) and (**B**) cleaved caspase-3 measured by Western blot densitometry. Representative Western blot images corresponding to each marker are shown above the graphs. Both markers were significantly increased in the control IgG group after I/R injury, and these increases were significantly suppressed by α-HMGB1 treatment. The Sham group (*n* = 1) is shown as a reference only; no statistical comparisons were performed due to the limited sample size. Values represent means ± SEM. 4-HNE, 4-hydroxynonenal; HMGB1, high mobility group box 1; I/R, ischemia–reperfusion. * Statistical significance: * *p* < 0.05, *** *p* < 0.001.

**Figure 8 ijms-26-08643-f008:**
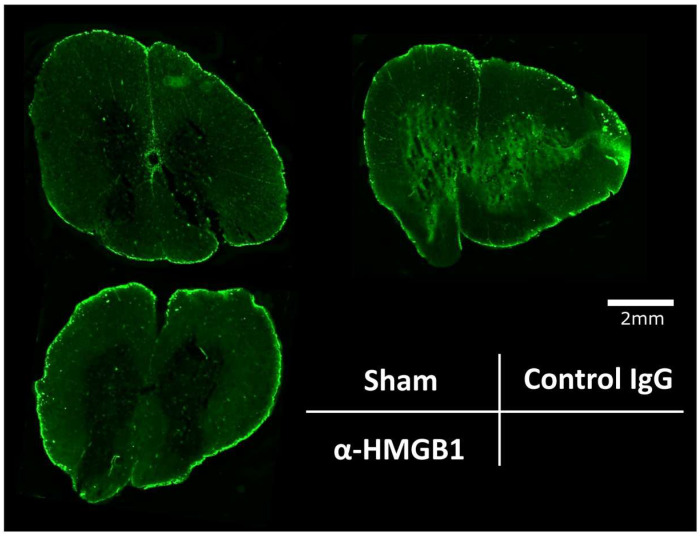
Effects of anti-HMGB1 monoclonal antibody on BSCB permeability at 48 h after I/R injury. Representative spinal cord sections immunostained with anti-albumin antibody. Images show albumin leakage predominantly in the gray matter. Scale bar = 2 mm. α-HMGB1, anti-HMGB1 monoclonal antibody; HMGB1, high mobility group box 1; I/R, ischemia–reperfusion.

**Figure 9 ijms-26-08643-f009:**
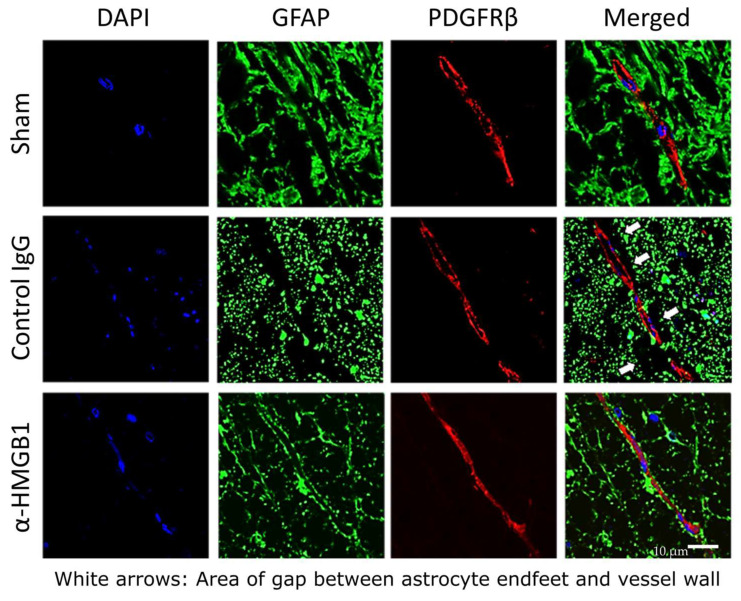
Structural alterations of the BSCB in the injury region visualized by double immunofluorescence staining. Representative horizontal sections of rabbit spinal cord double-stained for GFAP (astrocytes, green) and PDGFRβ-positive pericytes and astrocytic processes, indicative of astrocytic swelling and endfoot detachment. White arrows indicate detachment of astrocyte endfeet from vessel wall. DAPI (blue) was used for nuclear staining. Scale bar = 10 μm. α-HMGB1, anti-HMGB1 monoclonal antibody; BSCB, blood–spinal cord barrier; DAPI, 4′,6-diamidino-2-phenylindole; GFAP, glial fibrillary acidic protein; HMGB1, high mobility group box 1; PDGFRβ, platelet-derived growth factor receptor β; SCI/R, spinal cord ischemia–reperfusion.

**Figure 10 ijms-26-08643-f010:**
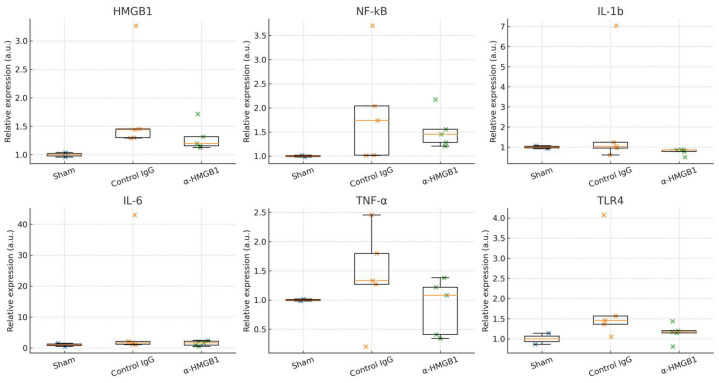
Quantitative real-time PCR analysis of inflammation-related gene expression in the gray matter of rabbit spinal cord 48 h after I/R injury. Expression levels of HMGB1, NF-κB, IL-1β, IL-6, TNF-α, and TLR4 in the Sham group (*n* = 2), control IgG group (*n* = 5), and α-HMGB1 group (*n* = 5). Data are presented as individual data points overlaid on boxplots (median and interquartile range). Statistical comparisons (Mann–Whitney U test) revealed no significant differences between the control IgG and α-HMGB1 groups (all *p* > 0.05). α-HMGB1, anti-HMGB1 monoclonal antibody; HMGB1, high mobility group box 1; IL, interleukin; I/R, ischemia–reperfusion; NF-κB, nuclear factor κB; TLR4, Toll-like receptor 4; TNF-α, tumor necrosis factor α.

**Table 1 ijms-26-08643-t001:** Clamp time-dependent variation in neurologic outcomes following SCI/R injury. I/R, ischemia–reperfusion. SCI/R, spinal cord ischemia–reperfusion.

Clamp Time (min)	Modified Tarlov Score After I/R		
	6 h After I/R	24 h After I/R	48 h After I/R
**30**	0	0	0
**20**	0	0	0
**15**	0	0	0
**12**	0	0	0
**11**	4	3 ± 1	1.5 ± 0.5
**10**	4	4	4
**9**	4	4	4
**8**	4	4	4

**Table 2 ijms-26-08643-t002:** Interoperative parameters.

	SpO_2_ (%)	HR (bpm)	BT (°C)	SBP (mmHg)	DBP (mmHg)
** *Pre-ischemia* **					
**Sham**	100	284	38.0	54	40
**Control IgG**	100	270.4 ± 11.2	38.4 ± 0.2	73.8 ± 3.5	49.0 ± 7.3
**α-HMGB1**	100	278.0 ± 4.9	38.3 ± 0.1	66.0 ± 5.8	52.2 ± 3.1
** *Post-ischemia* **					
**Sham**	100	259	37.4	36	30
**Control IgG**	99.8 ± 0.2	250.2 ± 6.8	37.6 ± 0.7	55.8 ± 0.3	42.0 ± 0.8
**α-HMGB1**	100	249.2 ± 7.0	37.5 ± 0.1	62.8 ± 5.2	43.0 ± 4.0

## Data Availability

The data presented in this study are available in Appendix A.

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
