# Peer review of "Anti-HMGB1 Antibody Therapy Ameliorates Spinal Cord Ischemia–Reperfusion Injury in Rabbits"

_ijms, 2025, doi:10.3390/ijms26178643_

Round 1

Reviewer 1 Report

Comments and Suggestions for Authors

Several changes must be included in the final version of the manuscript:

INTRODUCTION

  1. Authors should include information on the incidence and prevalence of spinal cord ischemia-reperfusion injury. Simply reporting overall spinal cord injury rates is not sufficient. Epidemiological information on thoracoabdominal aortic aneurysms should also be included.
  2. In the introduction, authors should include more details about the pathophysiological processes that occur in the spinal cord after ischemia-reperfusion, especially highlighting those that are different from a traumatic spinal cord injury (e.g., compression, contusion, laceration, section).
  3. There are publications in the scientific literature demonstrating the therapeutic effect of using antibodies against High Mobility Group Box 1 after spinal cord injury (e.g., Neurosci Res. 2021 Nov:172:13-25; Neurosci Res. 2019 Apr;141:63-70; Stem Cells. 2018 May;36(5):737-750; J Neurotrauma. 2019 Feb 1;36(3):421-435; Mol Med Rep. 2020 Dec;22(6):4725-4733; Neurosci Res. 2019 Apr;141:63-70; Stem Cells. 2018 May;36(5):737-750; Brain Res. 2017 Mar 15;1659:113-120; J Neurotrauma. 2023 Dec;40(23-24):2522-2540). What is new about this study compared to these existing publications? This information is highly relevant and should be included in the final version of the manuscript.

MATERIALS AND METHODS

  1. Authors should include information on the suppliers and references of the products used in the various procedures in this methodology section.
  2. What anatomical landmarks were used for the extraction of spinal cord segments from L5 to L7? Include this relevant information in the final version of the manuscript.
  3. How did the authors separately identify the spinal cord segments L5, L6, and L7? Include this information in the final version of the manuscript.
  4. What equipment was used to perform the histological sections of the spinal cord? Include information about the equipment, model, and supplier.
  5. What dehydration procedure was used to make the paraffin blocks? How was this procedure performed? Explain the entire procedure in more detail.
  6. Describe the histological procedure for hematoxylin-eosin staining in more detail. Indicate the brand and supplier of the hematoxylin-eosin used.
  7. How was the neuron count performed? Was a computer program (e.g., Image-J) used to perform this neuron count? How many sections per experimental group were analyzed? Include all this information in the final version of the manuscript.
  8. In the immunohistochemistry section, authors should include information on the antibody titer used (e.g. anti-HMGB1 1:200), and this for each of the antibodies used. The same applies to secondary antibodies. This information is also missing from the Western blot section.
  9. Authors should include information on the model, brand, and supplier of the microscopes (optical, fluorescence) used to visualize the spinal cord.
  10. In this section, the different experimental groups and the number of animals used in each experimental group should be made clearer.

RESULTS

  1. Figure 1 does not clearly show significant differences between the different experimental groups at the different study times. Does the IgG control group show significant differences with the Sham group? Does the IgG control group show significant differences with the HMGB1 group? Does the IgG control group show significant differences with the HMGB1 group? Please include all this information in the final version of the manuscript. The degree of significance of the various asterisks should also be indicated in the figure.
  2. In Figure 2, are there any differences between the IgG and HMGB1 groups compared to the Sham group? Please show them in the figure if there are any differences. If there are no differences, please indicate them in the text or on the figure itself.
  3. Figure 3 shows histological sections of the spinal cord in the left column, with a box indicating what was captured in the middle column. The right column shows high-magnification sections where motor neurons can be seen. The authors should use a box to indicate which area of the images in the middle column corresponds to the images in the right column. It would also be helpful for less experienced readers to mark the motor neurons with arrowheads.
  4. What criteria do the authors use to determine whether a motor neuron is viable or nonviable? Please include this information in the methodology.
  5. Figure 4C: What relevant information does this figure provide? Are there significant differences between the different experimental groups at each level of the spinal cord? The paragraph between lines 171 and 178 could be included in the legend for Figure 4. The authors should improve the wording of the results in Figure 4.
  6. In Figure 5A, there are very few cells with double labeling of Iba1 and HMGB1. To what cell type does the HMGB1 labeling correspond? Please include this information in the final version of the manuscript. The paragraph between lines 203-212 should be included in the legend for Figure 5. The authors should improve the wording of the results in Figure 5.
  7. Please include arrowheads in Figure 6A to indicate MPO-positive cells. Paragraphs 217-221 should be included in the legend for Figure 6, and the results from this figure should be better described in the text.
  8. The images in Figures 5 and 6, in which region of the spinal cord were they obtained? In the dorsal, ventral, or middle area of the spinal cord? Close to or far from the injury site? Please include this information in the final version of the manuscript.
  9. In the histograms in Figures 5, 6, and 7, there are significant differences between the IgG and HMGB1 groups compared to the Sham group. I think there are differences, but they are not indicated. Please include this information in the final version of the manuscript.
  10. Paragraph 240-244 should go in the legend for Figure 7. The results of Figure 7 must be described in the text.
  11. Figure 8 is missing the calibration bar. Were all images taken at the same magnification? Please include this information in the final version of the manuscript.
  12. Paragraph 270-279 should be placed in the legend of Figure 9.
  13. The histograms in Figure 10 show no significant differences with the Sham group. Are there really no differences with this experimental group? Please include this information in the final version of the manuscript.
  14. Paragraph 292-296 should be included in the legend of Figure 10.

DISCUSSION

  1. The authors should further discuss the intracellular pathways involved in HMGB1 activation in neurons, glial cells, and immune cells in the spinal cord injured by ischemia and reperfusion.
  2. The pathophysiology of spinal cord ischemia-reperfusion injury needs to be discussed in greater depth, and this pathophysiology should be related to HMGB1 activation.
  3. Authors should include and discuss epidemiological data on postoperative paraplegia following TAAA surgery. Is this type of paraplegia more common than paraplegia following spinal cord contusion or spinal cord section? Please discuss these aspects in more depth.
  4. Does anti-HMGB1 treatment have side effects? What would be the pros and cons of using this treatment in humans with spinal cord injury following ischemia and reperfusion? Are there clinical data on the use of this therapy in nervous system injuries? Please include information on these points in the final version of the manuscript.
  5. The authors should discuss the preclinical and clinical relevance of the manuscript's results.

Author Response

Answers for Reviewer 1

We sincerely thank you for taking the time to review our manuscript. We greatly appreciate your valuable comments and suggestions. Please find our responses below.

The Discussion section has been significantly revised and expanded. 

INTRODUCTION

l  Authors should include information on the incidence and prevalence of spinal cord ischemia-reperfusion injury. Simply reporting overall spinal cord injury rates is not sufficient. Epidemiological information on thoracoabdominal aortic aneurysms should also be included.

Answer: 

     Thank you very much for your important comments. The incidence of SCI after TAAA surgery is 3.3% according to a recent systematic review and meta-analysis, 4.0% for open surgery, and 2.9% for endovascular treatment. We also re-examined the frequency of delayed SCI. While the prevailing opinion in the past was around 30%, it is now suspected that this figure may be significantly underestimated, as patients are typically sedated upon returning to the postoperative intensive care unit. It is now suggested that the actual rate may exceed 80%. We have revised References 3 and 4 and modified the first paragraph of the Introduction to include specific frequency data.

(Before)Despite current preventive strategies, SCI occurs in up to 30% of patients undergoing these surgical treatments [3,4].

Reference:

3.         Etz, C.D.; Weigang, E.; Hartert, M.; Lonn, L.; Mestres, C.A.; Di Bartolomeo, R.; Bachet, J.E.; Carrel, T.P.; Grabenwoger, M.; Schepens, M.A.A.M.; et al. Contemporary spinal cord protection during thoracic and thoracoabdominal aortic sur-gery and endovascular aortic repair: a position paper of the vascular domain of the European Association for Ca-dio-Thoracic Surgery. Eur J Cardiothorac Surg. 2015, 47, 943-57.

4.         Jacobs, M.J.; Schurink, G.W.; Mees, B.M. Spinal Cord Ischaemia after Complex Procedures. Eur J Vasc Endovasc Surg. 2016, 52, 279-280.

(After) Page 1,Line44 – Page 2, Line 2

Permanent lower extremity paralysis due to SCI has been reported in 3.3% of patients undergoing treatment for TAAA, with an incidence of 4.0% following open surgery and 2.9% after endovascular repair [3]. Moreover, it has been suggested that the pro-portion of delayed-onset spinal cord injury among all SCI cases after TAAA repair, whether open or endovascular, may be unexpectedly high, potentially exceeding 80% [4].

Reference:

3.         Alzghari, T; An, K.R; Harik, L,; Rahouma, M.; Dimagli, A.; Perezgorvas-Olaria, R.; Demetres, M. Cancelli, G.; Soletti Jr, G.; Lau, C.; Girardi, L.N.; Gaudino, M. Spinal cord injury after open and endovascular repair of descending thoracic aneurysm and thoracoabdominal aortic aneurysm: an updated systematic review and meta-analysis. Ann Cardiothorac Surg 2023;21:409-417.

4.         Etz, C.D.; Weigang, E.; Hartert, M.; Lonn, L.; Mestres, C.A.; Di Bartolomeo, R.; Bachet, J.E.; Carrel, T.P.; Grabenwoger, M.; Schepens, M.A.A.M.; et al. Contemporary spinal cord protection during thoracic and thoracoabdominal aortic sur-gery and endovascular aortic repair: a position paper of the vascular domain of the European Association for Ca-dio-Thoracic Surgery. Eur J Cardiothorac Surg. 2015, 47, 943-57.

l  In the introduction, authors should include more details about the pathophysiological processes that occur in the spinal cord after ischemia-reperfusion, especially highlighting those that are different from a traumatic spinal cord injury (e.g., compression, contusion, laceration, section).

Answer: Thank you very much for your valuable comments. I have added a description about the differences between the pathophysiological processes of traumatic spinal cord injury and spinal cord disorders due to I/R. I felt that it did not fit well in the introduction, so I have incorporated it into the first paragraph of the discussion. If you feel that it would be more appropriate in the introduction or other sections, please let me know and I will revise it. (Page 14, Line 374 – Page 15, Line 416)

l  There are publications in the scientific literature demonstrating the therapeutic effect of using antibodies against High Mobility Group Box 1 after spinal cord injury (e.g., Neurosci Res. 2021 Nov:172:13-25; Neurosci Res. 2019 Apr;141:63-70; Stem Cells. 2018 May;36(5):737-750; J Neurotrauma. 2019 Feb 1;36(3):421-435; Mol Med Rep. 2020 Dec;22(6):4725-4733; Neurosci Res. 2019 Apr;141:63-70; Stem Cells. 2018 May;36(5):737-750; Brain Res. 2017 Mar 15;1659:113-120; J Neurotrauma. 2023 Dec;40(23-24):2522-2540). What is new about this study compared to these existing publications? This information is highly relevant and should be included in the final version of the manuscript.

Answer:  Thank you for your valuable comment. We agree that the description of our study objective was somewhat ambiguous. 

In contrast to previous studies that primarily used rodent models of traumatic or acute SCI, our work establishes a rabbit model of delayed spinal cord ischemia–reperfusion injury after TAAA surgery, which more faithfully reflects the clinical condition in humans. This unique model allowed us to evaluate the preventive potential of anti-HMGB1 mAb specifically against delayed-onset SCI. Thus, the novelty of our study lies in demonstrating the proof-of-concept of antibody therapy in a clinically relevant large-animal model, bridging preclinical findings toward surgical practice.

We have revised the statement to convey our objective more precisely.

(Before)

In this study, we investigated the efficacy of anti-HMGB1 mAb in mitigating spinal cord I/R injury in a rabbit model. Our primary focus was on its impact on the blood-spinal cord barrier (BSCB) and inflammatory responses. Additionally, we as-sessed the antibody’s effectiveness in managing neurological dysfunction. This study aims to elucidate the preventive utility of anti-HMGB1 mAb as a novel neuroprotective strategy for patients at risk of spinal cord injury during thoracoabdominal aortic aneu-rysm (TAAA) repair.

(After)Page 2, Line 70 - 75

In this study, we established a rabbit model of delayed spinal cord ischemia-reperfusion (I/R) injury and evaluated the efficacy of an anti-HMGB1 monoclonal antibody (mAb) in preventing this condition. We focused primarily on its effects on the blood–spinal cord barrier (BSCB) and inflammatory responses, and further assessed its ability to ameliorate neurological dysfunction. Collectively, these investigations aim to clarify the preventive potential of anti-HMGB1 mAb as a novel neuroprotective strategy for patients at risk of spinal cord injury during thoracoabdominal aortic aneurysm (TAAA) repair.

In addition, we have also made minor revisions to the conclusion to ensure consistency with the revised objective.

(Before) 

We established an optimized rabbit model of spinal cord I/R injury using an 11-minute occlusion protocol, which effectively replicates delayed-onset paraplegia or paraparesis observed after TAAA surgery. Using this model, we demonstrated that administration of anti-HMGB1 mAb significantly ameliorates spinal cord I/R injury. This therapeutic effect appears to be mediated through its neuroprotective, an-ti-inflammatory, and antioxidant properties, as well as its capacity to preserve BSCB integrity. These findings suggest that anti-HMGB1 mAb represents a promising ther-apeutic strategy for mitigating spinal cord injury associated with TAAA repair.

(After) Page 22 Line 776-786

We established a rabbit model of delayed SCI/R injury using an optimized 11-min occlusion protocol, which faithfully replicated the delayed-onset paraplegia observed after TAAA surgery. Using this model, we demonstrated that administration of an an-ti-HMGB1 mAb significantly ameliorated delayed SCI/R injury. The protective effects appeared to be mediated through the neuroprotective, anti-inflammatory, and antiox-idant properties of the antibody as well as its capacity to preserve BSCB integrity. Im-portantly, this exploratory proof-of-concept study provides preclinical evidence sup-porting anti-HMGB1 mAb administration as a potential preventive or therapeutic strategy for delayed spinal cord injury after aortic surgery, thereby underscoring its translational relevance as a bridge toward future efficacy and safety studies in humans.

MATERIALS AND METHODS

l  Authors should include information on the suppliers and references of the products used in the various procedures in this methodology section.

Answer: Thank you very much for your question. We have added the information in the methodology section. 

l  What anatomical landmarks were used for the extraction of spinal cord segments from L5 to L7? Include this relevant information in the final version of the manuscript.

Answer: Thank you very much for your question.  In addition to this question, we received other questions related to H&E staining, so we have significantly revised and added information on paraffin section preparation methods, dehydration methods, section preparation equipment, H&E staining methods, and motor neuron counting methods to the section titled “4.3. Hematoxylin and Eosin Staining.”

(After) 

Page 20 Line 666-674

For spinal cord harvesting, the abdominal cavity was opened to approach the anterior aspect of the spine. The L7 vertebra, located directly above the pelvis, was identified, and three consecutive vertebrae together with the enclosed spinal cord were removed. The extracted vertebrae were subsequently dissected from the posterior side to access the spinal canal, and the spinal cord was exposed. The cord was then transected be-tween vertebrae, and segments corresponding to the L5, L6, and L7 levels were care-fully harvested. Tissue samples from the L5, L6, and L7 spinal cord segments were partially preserved in 10 % formalin for histological evaluation, and the remaining por-tions were homogenized for western blot and PCR analyses.

l  How did the authors separately identify the spinal cord segments L5, L6, and L7? Include this information in the final version of the manuscript.

Answer: Thank you very much for your question. The answer is as stated in the question above.( Page 20, Line 666 – Page 20, Line 674)

l  What equipment was used to perform the histological sections of the spinal cord? Include information about the equipment, model, and supplier.

Answer: Thank you very much for your question. Axial sections were cut at 5 μm thickness using a rotary microtome (HM325, Thermo Fisher Scientific, Waltham, MA, USA). We added the information. (Page 20, Line 684 – Page 20, Line 685)

l  What dehydration procedure was used to make the paraffin blocks? How was this procedure performed? Explain the entire procedure in more detail.

Answer: Thank you very much for your question. The answer is as stated in the question above.( Page 19, Line 633 – Page 20, Line 653)

(After)

Axial sections were cut at 5 μm thickness using a rotary microtome (HM325, Thermo Fisher Scientific, Waltham, MA, USA).

l  Describe the histological procedure for hematoxylin-eosin staining in more detail. Indicate the brand and supplier of the hematoxylin-eosin used.

Answer: Thank you very much for your question. We have described more detail about H-E staining. (Page 20, Line 676 – Page 20, Line 699)

(After)

4.3. Hematoxylin and Eosin Staining

Formalin-fixed L5, L6, and L7 segment spinal cord tissues were thoroughly washed in PBS for 24 h with several buffer changes to remove residual fixative. Sam-ples were then dehydrated through a graded ethanol series (50%, 70%, 80%, 90%, 95%, and 100% × 3; ~4 h each), cleared in toluene (two changes, ~4 h each), and infiltrated with paraffin wax at 60 °C. Paraffin infiltration was performed stepwise, beginning with a toluene–paraffin mixture (1:1, 2 h), followed by two changes of pure paraffin (4 h each). Finally, tissues were embedded in pre-warmed paraffin molds and rapidly cooled at room temperature. Axial sections were cut at 5 μm thickness using a rotary microtome (HM325, Thermo Fisher Scientific, Waltham, MA, USA).

Hematoxylin–eosin staining was performed at the Central Research Laboratory, Okayama University, using their standard protocol. Briefly, paraffin sections were de-paraffinized in xylene, rehydrated through a graded ethanol series, stained with he-matoxylin, rinsed and blued in running tap water, counterstained with eosin, dehy-drated in graded ethanol, cleared in xylene, and mounted with a coverslip.

Spinal cord morphology was examined using LSM 780 confocal microscope (Carl Zeiss Inc., Thornwood, NY, USA), and injured neurons were identified by the presence of eosinophilic cytoplasm, pyknotic nuclei, and the reduced presence or absence of Nissl bodies. Normal neurons exhibited a polygonal shape with prominent Nissl bod-ies. Pathological changes in Nissl bodies were defined as loss of clear boundaries, de-creased staining intensity, and reduction in size compared with those of normal neu-rons.One transverse section per spinal cord level (L5, L6, and L7) was prepared, and all viable motor neurons located within the bilateral anterior horns of each section were counted and included in the analysis.

l  How was the neuron count performed? Was a computer program (e.g., Image-J) used to perform this neuron count? How many sections per experimental group were analyzed? Include all this information in the final version of the manuscript.

Answer: Thank you very much for your question. For motor neuron counts, a single tissue section was prepared at each spinal cord level (L5, L6, and L7) at approximately equal intervals, and all motor neurons in the bilateral anterior horn of that section were visually counted. In addition, we would like to note that paraffin sections and slides were prepared from three spinal cord levels (L5, L6, and L7). The previous description unintentionally gave the impression that only L6 was processed, and this has now been corrected for accuracy. We have added the information as described above (Page 20 Line 691 – Page 20 Line 699).

l  In the immunohistochemistry section, authors should include information on the antibody titer used (e.g. anti-HMGB1 1:200), and this for each of the antibodies used. The same applies to secondary antibodies. This information is also missing from the Western blot section.

Answer: Thank you very much for your question. All details regarding dilution ratios have been added to sections “4.4. Immunofluorescence Staining” and “4.5. Western Blot Analysis”. Please refer to the main text for details.

l  Authors should include information on the model, brand, and supplier of the microscopes (optical, fluorescence) used to visualize the spinal cord.

Answer: Thank you very much for your question. We use a confocal laser microscope LSM780 (Carl Zeiss Inc., Thornwood, NY, USA). Please refer to sections “4.3. Hematoxylin and Eosin Staining” and “4.4. Immunofluorescence Staining.” 

l  In this section, the different experimental groups and the number of animals used in each experimental group should be made clearer.

Answer: Thank you very much for the question. The numbers of samples used for each figure (Sham – Control – Anti-HMGB1 mAb) are as follows:

Fig. 1: Tarlov score, 2 – 5 – 5

Fig. 2: ELISA Plasma HMGB1, 2 – 5 – 5

Fig. 4: Neuronal counts, 1 – 5 – 5

Fig. 5B:  HMGB1 positive cell area, 2 – 5 – 5

Fig. 5C:  Microglia count, 2 – 5 – 5

Fig. 6B: Neutrophils count, 2 – 5 – 5

Fig. 7A: Western Blot:4-HNE , 1 – 5 – 5

Fig. 7B: Western Blot: caspase3, 1 – 5 – 5

Fig. 10:  PCR 2 – 5 – 5 for each graphs.

Because the amount of spinal cord tissue available from each rabbit was limited, we made every effort to allocate samples so that as many analyses as possible could be performed across all groups. Nevertheless, in some instances, assays could not be completed in the sham group. Importantly, we believe that this limitation did not materially affect the overall results or conclusions of the present study. Through this revision, we were also able to clarify and explain these points more explicitly.

We sincerely appreciate your appropriate and constructive comment.

We have added an explanation regarding the handling of the sham group in the “4.8 Statistical Analysis”. There were also questions about statistical methods, so we have significantly revised the “4.8. Statistical Analysis” section as follows.

(After) 

4.8. Statistical Analysis 

Continuous variables were represented as the mean ± standard error of the mean (SEM). Intraoperative blood pressure, heart rate, SpO₂, body temperature values, pre- and post-operative modified Tarlov score, and the results of quantitative real-time PCR were compared between the control IgG group and the anti-HMGB1 mAb group using the exact Mann–Whitney U test. An unpaired t-test was used to analyze serum HMGB1 concentration, the number of viable motor neurons, the number of microglia per animal, the number of MPO-positive neutrophils, and the relative intensities of caspase-3 and 4-HNE. The significance level was set at p < 0.05. To complement hypothesis testing, effect sizes were estimated using Cliff’s delta with 95% bootstrap confidence intervals, and Hodges–Lehmann median differences were also calculated. Outliers in the PCR dataset were identified a priori using the interquartile range (IQR) method (values < Q1 – 1.5 × IQR or > Q3 + 1.5 × IQR), but all values were retained in the primary analysis; sensitivity analyses excluding IQR-flagged values are presented in the Supplementary Data.

Although a Sham group was included for reference, statistical comparisons with the Sham group were not performed because the sample size was only 1–2 animals and any differences from the control or anti-HMGB1 mAb groups were visually evident; thus, such comparisons were considered of limited statistical value. In principle, two animals were assigned to the Sham group; however, in some experiments (Figures 4, 7A, and 7B), only one sample was available due to limited tissue availability during har-vesting. 

RESULTS

l  Figure 1 does not clearly show significant differences between the different experimental groups at the different study times. Does the IgG control group show significant differences with the Sham group? Does the IgG control group show significant differences with the HMGB1 group? Does the IgG control group show significant differences with the HMGB1 group? Please include all this information in the final version of the manuscript. The degree of significance of the various asterisks should also be indicated in the figure.

Answer: Thank you very much for your comment. As mentioned above, the sham group was treated as reference values and was not included as a comparison group for statistical analyses in the present study.

In Figure 1, the asterisks indicate significant differences between the control and anti-HMGB1 mAb groups at 24 hours and 48 hours. To improve clarity, we have revised Figure 1 accordingly. In addition, we have revised the manuscript to clarify that the sham group was not used as a comparison group. We apologize for the confusion caused by the previous wording, which may have implied otherwise.

Currently, we are using the Mann–Whitney U test, but we also performed a two-way ANOVA with time and anti-HMGB1 mAb treatment as factors. This analysis confirmed significant differences between the two groups at 24 hours (P = 0.0339) and 48 hours (P = 0.0337). Both time (P = 0.0003) and treatment (P = 0.0053) were significant factors, and there was no interaction between time and treatment (P = 0.1486).

If you would prefer that we present the statistical results based on the two-way ANOVA, we would be pleased to revise the manuscript accordingly at your request.

(Figure revision) Figure 1 has been updated.  In addition, we have added the relevant observation as a footnote. 

(Before) 

Compared to baseline values of the Sham group, neurological deficits were observed in all rabbits at six, 24, and 48 hours after I/R injury in the control IgG group. Rabbits treated with anti-HMGB1 mAb exhibited significant improvement at 24 and 48 hours after I/R injury from the control IgG group.

(After) 

). In the control IgG group, clear neurological deficits developed after I/R injury and showed a progressive worsening at 24 and 48 h. Treatment with anti-HMGB1 mAb significantly ameliorated these deficits compared with the control IgG group; however, the recovery did not reach the intact level observed in the Sham group. Notably, rabbits in the Sham group exhibited no neurological deficits throughout the observation period.

l  In Figure 2, are there any differences between the IgG and HMGB1 groups compared to the Sham group? Please show them in the figure if there are any differences. If there are no differences, please indicate them in the text or on the figure itself.

Answer: Thank you very much for your comment. As mentioned above, the sham group was treated as reference values and was not included as a comparison group for statistical analyses in the present study. We have added a comment regarding the sham group and also included a description of the results in the footnote of Figure 2. Regarding the treatment of the sham group, we repeatedly referred to it in the text and in the footnotes of each figure to avoid misunderstanding by readers.

(Before) 

no comment about the Sham group.

(After)  Page 3, Line 116-118

In the Sham group, although a slight increase was observed, the values remained nearly equivalent to the preoperative level, and no substantial changes were detected (Figure 2).

l  Figure 3 shows histological sections of the spinal cord in the left column, with a box indicating what was captured in the middle column. The right column shows high-magnification sections where motor neurons can be seen. The authors should use a box to indicate which area of the images in the middle column corresponds to the images in the right column. It would also be helpful for less experienced readers to mark the motor neurons with arrowheads.

Answer:   Thank you very much for your valuable advice. In accordance with your suggestion, we have added arrows to the figure.  Regarding the boxed areas, we would like to clarify that the high-magnification images shown in the right column were obtained to illustrate the representative morphology of Nissl bodies, rather than being direct enlargements of the corresponding boxed regions in the middle column. Therefore, not all images in the right column correspond exactly to the boxed areas in the middle column. As this study has already completed the target exploration phase, if strict correspondence between the boxed regions and the high-magnification images is required, we would need additional time to identify the relevant specimens and perform re-imaging. We kindly ask for your guidance as to whether such correspondence is mandatory.

(Figure revision) Figure 3 and its footnote have been updated.

l  What criteria do the authors use to determine whether a motor neuron is viable or nonviable? Please include this information in the methodology.

Answer: Thank you very much for your question. Normal motor neurons exhibit large, darkly stained, polygonal structures with well-defined Nissl bodies. As the Tarlov score increases, pathological changes are observed, including blurred boundaries, paler staining, and reduction in the size of Nissl bodies. In Figure 3, the representative images presented for the Sham and Tarlov 4 groups show mostly normal morphology, and the Tarlov 3 group also appears largely normal, although subtle changes can be observed in the selected images (which may naturally lead to some variation in counting depending on the observer’s judgment). In contrast, in the Tarlov 2 and 1 groups, the majority of motor neurons are clearly non-viable, and thus their classification is generally consistent regardless of the observer. While individual variability in counting cannot be entirely eliminated, as shown in the images, viable and non-viable neurons can be readily distinguished, and we believe this minimizes any potential bias. For this reason, we provided the representative high-magnification images in the right column of Figure 3. We have also added clarifications to the Methods, Results, and the legend of Figure 3 to address this point.

(Result: Before)

Representative coronal sections fixed 48 h after I/R injury and stained with hema-toxylin and eosin are shown in Figure 3. In the anti-HMGB1 mAb group with a Tarlov score of 4, numerous viable motor neurons exhibited polygonal structures and Nissl bodies, which are characterized by basophilic staining in the cytoplasm (equivalent to the Sham group level). Conversely, as the Tarlov score decreased, the number of viable motor neurons in the anterior horn also diminished. In the control IgG group with a Tarlov score of 1, imaging revealed extensive infarct lesions with impaired motor neu-rons, which were characterized by eosinophilic cytoplasmic dye and vacuolization.

(Result: After)Page 6, Line 156 - 166

Representative coronal sections fixed 48 h after I/R injury and stained with hema-toxylin and eosin are shown in Figure 3. In the anti-HMGB1 mAb group with a Tarlov score of 4, numerous viable motor neurons exhibited polygonal structures and Nissl bodies, which are characterized by basophilic staining in the cytoplasm (equivalent to the Sham group level). Conversely, as the Tarlov score decreased, the number of viable motor neurons in the anterior horn also diminished. Along with this decline, the mor-phology of Nissl bodies showed progressive alterations: from large, darkly stained structures with well-defined boundaries to smaller, paler bodies with indistinct mar-gins. In the control IgG group with a Tarlov score of 1, imaging revealed extensive in-farct lesions with impaired motor neurons, which were characterized by eosinophilic cytoplasmic dye and vacuolization.

(Figure 3 foot note: Before) 

The pictures on the left column are the whole images of the spinal cord tissues. The pictures of the middle column are the anterior horn region with various infarction are-as with each Tarlov score. The pictures of right column are magnifications of motor neurons with various degree of damages with each Tarlov score. α-HMGB1, anti-HMGB1 monoclonal antibody; HMGB-1, human mobility group box 1; I/R, ischemia-reperfusion. 

(Figure 3 footnote: After)

The pictures in the left column show whole images of the spinal cord tissues. The boxed areas in the left column are magnified in the middle column, which depicts the anterior horn region with various infarction areas corresponding to each Tarlov score. The right column presents representative high-magnification images of motor neurons. Normal motor neurons exhibit large, darkly stained, polygonal structures with well-defined Nissl bodies (black arrows). As the Tarlov score decreases, pathological changes become evident, including blurred boundaries, paler staining, and reduction in the size of Nissl bodies. 

α-HMGB1, anti-HMGB1 monoclonal antibody; HMGB1, high mobility group box 1; I/R, ischemia-reperfusion.

(For Method) 

Changes to the Method section are as described in the explanation of the changes to the Method section that have already been made.

l  Figure 4C: What relevant information does this figure provide? Are there significant differences between the different experimental groups at each level of the spinal cord? The paragraph between lines 171 and 178 could be included in the legend for Figure 4. The authors should improve the wording of the results in Figure 4.

Answer: Thank you for your question. We have provided a more detailed explanation of the significance of Figure 4C, and we have also added a description of the findings in the footnote of Figure 4.

(Figure revision) We have added a comment regarding the sham group and also included a description of the results in the footnote of Figure 4.

(Text: Before revision) 

Furthermore, bar graphs depicting individual animal data at each spinal cord level (L5, L6, and L7) revealed a trend toward decreasing numbers of viable motor neurons at more distal levels. While a few animals demonstrated substantial preservation of mo-tor neurons at the L5 and L6 levels, and one such case was also observed at L7, these outlier cases became less frequent as the level moved distally. This pattern suggests a reduction in inter-individual variability in neuronal survival at more distal spinal cord levels (Figure 4C).

(Text: After revision) Page 6, Line 167-193

Furthermore, Figure 4C illustrates the number of viable motor neurons 48 h after SCI/R injury, assessed at three spinal cord levels (L5, L6, and L7). The Sham group (N = 1) ex-hibited no neurological deficits and maintained motor neuron counts comparable to the preoperative state; however, statistical comparisons with this group were not per-formed because of the limited sample size. In the control IgG and anti-HMGB1 mAb groups (N = 5 each), no statistically significant differences were detected at any spinal cord level. Notably, in both groups, a few animals displayed exceptionally high num-bers of surviving motor neurons, which could be regarded as outliers. These outliers likely accounted for the absence of statistical significance despite the apparent trend toward neuronal preservation in the anti-HMGB1 mAb group. Such cases are consid-ered to reflect inter-individual variability in the development of collateral circulation within the spinal cord. Supporting this interpretation, outliers were less frequently observed at the L7 level compared with L5 and L6, suggesting that collateral blood supply becomes progressively less effective at more distal levels from the aortic clamping site. This phenomenon is further explained by the anatomical fact that rab-bits possess fewer collateral pathways in the spinal cord than humans.

In addition, Figures 4A and 4B were difficult to understand, so I edited them to make them easier to understand.

(Before revision)

Figure 4 presents the analysis of viable motor neurons in the anterior horn of the spinal cord at the L5 to L7 levels, which are presumed to be affected by reduced blood flow following abdominal aortic clamping in this study. We first compared the number of viable motor neurons in a single transverse section of the anterior horn at the L7 level, which is the most distal and therefore expected to be the most severely affected by ischemia. Although this comparison did not reach statistical significance (P = 0.126), the Cliff’s delta value was +0.84, indicating a very strong effect size (Figure 4A). Based on this finding, we then quantified viable motor neurons in a single transverse section at each of the L5, L6, and L7 levels and compared the slice-based data. The an-ti-HMGB1 monoclonal antibody (mAb) group exhibited a significantly greater number of viable anterior horn motor neurons than the control IgG group (P < 0.05), suggesting a neuroprotective effect of the anti-HMGB1 antibody (Figure 4B).

(After revision) Page 6 Line 167-178

Figure 4 shows the analysis of viable motor neurons in the anterior horn of the spinal cord at the L5 to L7 levels, which were presumed to be affected by reduced blood flow following abdominal aortic clamping. We first compared the number of vi-able motor neurons in a single transverse section of the anterior horn at the L7 level, which is the most distal and therefore expected to be the most severely affected by is-chemia. Although this comparison did not reach statistical significance (P = 0.126), the Cliff’s delta value was +0.84, indicating a very strong effect size (Figure 4A). On the basis of this finding, we then quantified viable motor neurons in a single transverse section at the level of L5, L6, and L7 levels and analyzed the data on a per-animal basis. The anti-HMGB1 mAb group demonstrated a significantly greater total number of via-ble anterior horn motor neurons compared with the control IgG group (P < 0.05), indi-cating a neuroprotective effect of the anti-HMGB1 antibody (Figure 4B).

l  In Figure 5A, there are very few cells with double labeling of Iba1 and HMGB1. To what cell type does the HMGB1 labeling correspond? Please include this information in the final version of the manuscript. The paragraph between lines 203-212 should be included in the legend for Figure 5. The authors should improve the wording of the results in Figure 5.

Answer:Thank you for your insightful comment. As shown in Figure 5A, very few cells exhibited double labeling of Iba1 and HMGB1. We believe this observation can be explained by several factors. First, HMGB1 is a nuclear protein that is rapidly translocated and released extracellularly from neurons and astrocytes under stress or injury, whereas such pronounced nuclear-to-cytoplasmic translocation does not always occur in microglia. Because immunostaining detects intracellular HMGB1, microglia that had already released HMGB1 may appear weakly stained and thus rarely show double positivity. Second, microglia generally express relatively low levels of HMGB1 under physiological conditions. Following spinal cord ischemia, the transient upregulation of HMGB1 is thought to occur predominantly in injured neurons and reactive astrocytes, while microglia act primarily as responders through HMGB1 receptor signaling rather than as the major source of HMGB1. Third, the time point examined in this study (within 48 h after ischemia) may have captured the acute phase, when neurons strongly express and release HMGB1, whereas robust microglial HMGB1 expression is more likely to emerge during the subacute to chronic phases. Finally, technical aspects should also be considered: Iba1 is a cytoplasmic marker, whereas HMGB1 localizes to both the nucleus and cytoplasm. Differences in subcellular localization, morphology, and staining intensity may make the detection of colocalization more difficult.

The following corrections have been made.

(Before revision) To investigate the effects of anti-HMGB1 mAb on spinal cord inflammation, we examined intracellular HMGB1 translocation and retention, the number of microglial cells, and neutrophil infiltration using immunofluorescence staining in the anterior horn of the spinal cord following ischemia-reperfusion injury (Figure 5). The number of cells retaining HMGB1 was reduced in the control IgG group; this reduction was attenuated by anti-HMGB1 mAb treatment, resulting in levels comparable to those observed in the sham group. Furthermore, expression of Iba1—a specific marker for microglia—was elevated in the control IgG group but decreased in the anti-HMGB1 mAb group. Representative immunofluorescence images are shown in Figure 5A. Quantification of the area occupied by anti-HMGB1–positive cells and the number of microglia revealed statistically significant differences between the control IgG group and the an-ti-HMGB1 mAb group (Figure 5B, 5C). These findings suggest that spinal cord damage induced by ischemia is attenuated by administration of anti-HMGB1 mAb.

(After revision)Page 9, Line 232-245

To investigate the effects of the anti-HMGB1 mAb on spinal cord inflammation, we examined intracellular HMGB1 translocation and retention, the number of microglial cells, and neutrophil infiltration by immunofluorescence staining in the anterior horn of the spinal cord following I/R injury (Figure 5). The number of cells retaining HMGB1 was reduced in the control IgG group, and this reduction was attenuated by an-ti-HMGB1 mAb treatment, resulting in levels comparable to those observed in the Sham group. Furthermore, the expression of Iba1, a specific marker for microglia, was ele-vated in the control IgG group, but decreased in the anti-HMGB1 mAb group. Repre-sentative immunofluorescence images are shown in Figure 5A, in which only a few cells exhibit double labeling of Iba1 and HMGB1. Quantification of the area occupied by an-ti-HMGB1–positive cells and the number of microglia revealed statistically significant differences between the control IgG and anti-HMGB1 mAb groups (Figure 5B, 5C). These findings suggest that the spinal cord damage induced by ischemia is attenuated by the administration of anti-HMGB1 mAb.

In addition, we have added a paragraph about this microglia issue on the Discussion section.

(Before revision) 

No comment on the Discussion section.

(After revision) Page 17, Line 520-533

Regarding microglial responses, our immunofluorescence analysis revealed that only a few microglial cells exhibited double labeling of Iba1 and HMGB1 at 48 h after I/R injury. Several factors may account for this observation. First, HMGB1 is a nuclear protein that is rapidly translocated and released extracellularly from neurons and as-trocytes under stress or injury, whereas such pronounced nuclear-to-cytoplasmic translocation does not always occur in microglia. Second, microglia generally express relatively low basal levels of HMGB1, and they are often considered to function pri-marily as responders through HMGB1 receptor signaling rather than as major sources of HMGB1. Third, HMGB1 expression dynamics are time dependent; robust microglial HMGB1 expression may appear during subacute or chronic phases, whereas our anal-ysis was limited to the acute phase within 48 h. Finally, methodological aspects should be considered, as Iba1 is a cytoplasmic marker whereas HMGB1 localizes to both the nucleus and cytoplasm, making colocalization more difficult to detect. Taken together, these factors may explain why only a small number of Iba1/HMGB1 double-positive cells were observed in this study.

We also made some changes to Limitation with this in mind.

(Before revision)This study has several limitations beyond those previously discussed. First, alt-hough circulatory and respiratory parameters were stabilized as much as possible during surgery, as shown in Table 2, they were not completely uniform across all ani-mals. Second, the postoperative observation period was limited to 48 hours. It is possi-ble that delayed spinal cord injury could manifest, or conversely, that existing deficits could improve beyond this time point. The relevance of the 48-hour endpoint has been addressed earlier. Third, individual differences in collateral circulation among rabbits were not evaluated in this study, which may have influenced the extent of spinal cord ischemia and reperfusion injury.

(After revision) Page 19, Line 606-617

This study had several limitations beyond those previously discussed. First, alt-hough circulatory and respiratory parameters were stabilized as much as possible during surgery, as shown in Table 2, they were not completely uniform across all ani-mals. Second, the postoperative observation period was limited to 48 h. As a result, the results did not account for delayed spinal cord injury or improvements in the existing deficits that appear after 48 h. The relevance of the 48-h endpoint has been addressed earlier. Third, individual differences in collateral circulation among rabbits, which may have influenced the extent of spinal cord I/R injury, were not evaluated in this study. Fourth, only male rabbits were used in the experiments. Potential sex-related differ-ences in vascular physiology, inflammatory responses, and susceptibility to ische-mia-reperfusion injury were therefore not addressed, and future studies including both sexes will be necessary to assess the generalizability of the present findings.

l  Please include arrowheads in Figure 6A to indicate MPO-positive cells. Paragraphs 217-221 should be included in the legend for Figure 6, and the results from this figure should be better described in the text.

Answer:  Thank you very much for your question. I agree that it was unclear. I have added arrows to the MPO-positive neutrophils in Figure 6. I have also added a description of the findings to the footnote of Figure 6. The figure used in the initial submission had a vivid blue color, which made it difficult to recognize the cells to be counted, so I used a photo with a darker background. I think this will make it easier to clearly identify the cells to be counted.

 (Figure revision) Figure 6 has been updated. In addition, we have added some comments in Figure 6 footnote.

l  The images in Figures 5 and 6, in which region of the spinal cord were they obtained? In the dorsal, ventral, or middle area of the spinal cord? Close to or far from the injury site? Please include this information in the final version of the manuscript.

Answer: Thank you for your question. First, regarding the location, it is the anterior horn of L6. Motor neurons are the focus of this study, so we used tissue samples from L5, L6, and L7, but we only performed the experiment on the middle L6. The excess L5 and L7 tissue was used for tests that required homogenization. 

l  In the histograms in Figures 5, 6, and 7, there are significant differences between the IgG and HMGB1 groups compared to the Sham group. I think there are differences, but they are not indicated. Please include this information in the final version of the manuscript.

Answer: Thank you very much for your question. As mentioned above, please note that we have not conducted any comparisons with Sham. 

l  Paragraph 240-244 should go in the legend for Figure 7. The results of Figure 7 must be described in the text.

Answer: Thank you very much for your comment. Following your advice, I added an explanation to the footnote in Figure 7. I tried to keep it short and simple, rather than repeating the explanation in the main text. Furthermore, we have enhanced the main text explanations.

(Before)

At 48 hours after I/R, oxidative stress and cell death in the spinal cord tissue were assessed by measuring 4-hydroxynonenal (4-HNE) and cleaved caspase-3 levels using ELISA (Figure 7). The expression of cleaved caspase-3 was significantly elevated in the control IgG group compared to the sham group, whereas administration of an-ti-HMGB1 monoclonal antibody (mAb) markedly suppressed this increase (P < 0.05) (Figure 7A). Similarly, the expression of 4-HNE was significantly increased following I/R injury but was substantially attenuated by anti-HMGB1 mAb treatment when compared with the control IgG group (P < 0.001) (Figure 7B). These findings were fur-ther corroborated by Western blot analysis (Supplemental File S1).

(After)Page 11 Line 283-293

At 48 h after I/R, oxidative stress and neuronal apoptosis in the spinal cord were evaluated by quantifying 4-hydroxynonenal (4-HNE), a marker of lipid peroxidation, and cleaved caspase-3, a key executor of apoptosis, using western blot densitometry (Figure 7). Representative Western blot images of each marker are also shown above the graphs. The expression of 4-HNE was significantly elevated in the control IgG group compared with the Sham group, and this increase was significantly suppressed by an-ti-HMGB1 mAb treatment (P < 0.05) (Figure 7A). Similarly, cleaved caspase-3 expression markedly increased following I/R injury but was substantially attenuated by an-ti-HMGB1 mAb treatment (P < 0.001) (Figure 7B). The Sham group (n = 1) is shown as a reference only and was not included in the statistical analyses. Together, these results demonstrate that anti-HMGB1 mAb attenuates both oxidative stress and apoptosis after SCI/R injury.

l  Figure 8 is missing the calibration bar. Were all images taken at the same magnification? Please include this information in the final version of the manuscript.

Answer: Thank you very much for your question.  Figure 8 has been revised to include a scale bar.

l  Paragraph 270-279 should be placed in the legend of Figure 9.

Answer: Thank you very much for your advice. I have added that description to Figure 9 footnote.

l  The histograms in Figure 10 show no significant differences with the Sham group. Are there really no differences with this experimental group? Please include this information in the final version of the manuscript.

Answer: Thank you very much for your question. As mentioned above, please note that we have not conducted any comparisons with Sham.

l  Paragraph 292-296 should be included in the legend of Figure 10.

Answer: Thank you very much for your advice. I have added that description to Figure 10 footnote.

DISCUSSION

l  The authors should further discuss the intracellular pathways involved in HMGB1 activation in neurons, glial cells, and immune cells in the spinal cord injured by ischemia and reperfusion.

l  The pathophysiology of spinal cord ischemia-reperfusion injury needs to be discussed in greater depth, and this pathophysiology should be related to HMGB1 activation.

Answer: Thank you for your comments. I believe these two points are related, so I will respond to them together. I have made significant additions and revisions to the Discussion section. I have moved the summary of the overall intracellular reactions and pathophysiology to the first paragraph to facilitate understanding, and I have also significantly changed the structure to describe the main results of this study. 

 The changes to the Discussion section are extensive and voluminous, so I would appreciate it if you could review the main section of the Discussion in the main text.  (Page 14 Line 374-Page 15 Line 416)

l  Authors should include and discuss epidemiological data on postoperative paraplegia following TAAA surgery. Is this type of paraplegia more common than paraplegia following spinal cord contusion or spinal cord section? Please discuss these aspects in more depth.

Answer: Thank you for your comment. We have added a comment to the first paragraph of the Discussion section, which provides an overview of the background. (Page 15 Line 404 – 416)

l  Does anti-HMGB1 treatment have side effects? What would be the pros and cons of using this treatment in humans with spinal cord injury following ischemia and reperfusion? Are there clinical data on the use of this therapy in nervous system injuries? Please include information on these points in the final version of the manuscript. 

Answer: Since this is antibody therapy, the target is quite specific and there is a possibility that side effects can be suppressed, but it is necessary to carefully evaluate immunogenicity, infection, and the occurrence of disorders due to excessive HMGB1 suppression. Unfortunately, after researching, I believe there are no clinical trials for anti-HMGB1 antibodies. I have noted this in the Discussion section. (Page 18, Line 596 – 605)

l  The authors should discuss the preclinical and clinical relevance of the manuscript's results.

Answer: The significance of this study lies not in the discovery of a new mechanism, but in the fact that it is an exploratory study that has established the proof of concept for the use of anti-HMGB1 antibodies as a treatment or preventive measure for delayed spinal cord injury in cardiovascular surgery aortic surgery, using an established animal model with high relevance for future efficacy and safety testing. I believe that this study has great significance as a bridge study. I have added the above statement to the text added in response to the above question. (Page 18 Line 582-605)

Reviewer 2 Report

Comments and Suggestions for Authors

This manuscript presents a well-structured and thorough experimental study investigating the neuroprotective effect of an anti-HMGB1 monoclonal antibody (mAb) in a rabbit model of spinal cord ischemia-reperfusion (I/R) injury. The authors demonstrate that anti-HMGB1 mAb improves neurological outcomes, reduces inflammation and oxidative stress, and preserves blood-spinal cord barrier (BSCB) integrity. The study is timely and addresses a clinically relevant complication following thoracoabdominal aortic aneurysm (TAAA) surgery. The methodology is sound, and the authors appropriately discuss their findings in the context of existing literature.

However, there are several aspects that require clarification or improvement before the manuscript can be accepted for publication.

  1. Limited Sample Size

The study relies on small group sizes (n = 5 for experimental groups and n = 2 for the sham group), which limits statistical power. The conclusions should be presented with caution, and more emphasis should be placed on the exploratory nature of the findings.

  1. Inconsistency in Plasma HMGB1 Levels

Although spinal cord tissue showed decreased HMGB1 staining in the treatment group, plasma HMGB1 levels did not differ between groups. This discrepancy warrants further discussion or clarification. Could differences in sampling time or HMGB1 turnover explain this?

  1. PCR Data Handling and Exclusion of Outliers

Several outliers in the qPCR dataset were excluded using IQR criteria. Given the small sample size, such exclusions may bias the results. Were these exclusions pre-specified? A sensitivity analysis including all values should be considered or at least discussed.

  1. Lack of Long-Term Functional Outcome Assessment

The study endpoint was 48 hours post-reperfusion. Although the authors justify this based on previous literature, a longer follow-up (e.g., 7–14 days) would better represent delayed neuronal loss and functional recovery in translational terms.

  1. Gender Bias

Only male rabbits were used. While this is common in preclinical studies, the exclusion of females should be acknowledged as a limitation, particularly when studying neuroinflammatory processes that may be sex-dependent.

  1. Lack of Mechanistic Insight Beyond TLR4

The study focuses largely on HMGB1-TLR4 signaling. However, other HMGB1 receptors (e.g., RAGE) are known to contribute to I/R injury. Inclusion of additional markers or even speculative discussion of these alternative pathways would enhance the mechanistic depth.

  1. Some key references are missing, such as: doi: 10.1007/s12035-025-04794-9

  1. Figure Legends Are Too Brief

The figure legends are generally too short and lack detail. Each legend should fully describe what is shown, including the biological context, experimental groups, number of replicates, and statistical annotations. A single-sentence legend is insufficient.

  1. Typographical Error in Figure 4

There is a typographical error in the Y-axis label of Figure 4 (“motor Eurons”). This should be corrected to “motor neurons.”

  1. Figure 7 – Missing Representative Western Blot Images

Figure 7 presents quantification graphs for Western blot results and refers readers to raw data in the supplementary file, which is commendable. However, representative, well-labeled Western blot images should also be included in the main figure panel alongside the graphs to improve readability and transparency.

  1. Language and Style

The English language requires improvement throughout the manuscript. Although generally understandable, the text contains grammatical errors, awkward phrasing, and overly long sentences. Professional editing is strongly recommended to improve clarity, flow, and readability.

Author Response

Answers for Reviewer 2

We sincerely thank you for taking the time to review our manuscript. We greatly appreciate your valuable comments and suggestions. Please find our responses below.

The Discussion section has been significantly revised and expanded. For details, please refer to the original paper.

1.Limited Sample Size

The study relies on small group sizes (n = 5 for experimental groups and n = 2 for the sham group), which limits statistical power. The conclusions should be presented with caution, and more emphasis should be placed on the exploratory nature of the findings.

Answer:  

Thank you very much for your valuable comments. Added comments in Discussion and Conclusion that this study is exploratory in nature.

(Discussion) Please check, Page 18, Line 607 – Page 19 618

(Conclusions: Before)

We established a rabbit model of delayed SCI/R injury using an optimized 11-min occlusion protocol, which faithfully replicated the delayed-onset paraplegia or para-paresis observed after TAAA surgery. Using this model, we demonstrated that admin-istration of an anti-HMGB1 mAb significantly ameliorated delayed spinal cord I/R in-jury. The protective effects were appeared to be mediated through the neuroprotective, anti-inflammatory, and antioxidant properties of the anti-HMGB1 mAb as well as its capacity to preserve BSCB integrity. These findings suggest that anti-HMGB1 mAb administration represents a promising therapeutic strategy for mitigating spinal cord injury associated with TAAA repair.

(Conclusions: After)

We established a rabbit model of delayed SCI/R injury using an optimized 11-min occlusion protocol, which faithfully replicated the delayed-onset paraplegia observed after TAAA surgery. Using this model, we demonstrated that administration of an anti-HMGB1 mAb significantly ameliorated delayed SCI/R injury. The protective effects appeared to be mediated through the neuroprotective, anti-inflammatory, and antioxidant properties of the antibody as well as its capacity to preserve BSCB integrity. Importantly, this exploratory proof-of-concept study provides preclinical evidence sup-porting anti-HMGB1 mAb administration as a potential preventive or therapeutic strategy for delayed spinal cord injury after aortic surgery, thereby underscoring its translational relevance as a bridge toward future efficacy and safety studies in humans.

2.Inconsistency in Plasma HMGB1 Levels

Although spinal cord tissue showed decreased HMGB1 staining in the treatment group, plasma HMGB1 levels did not differ between groups. This discrepancy warrants further discussion or clarification. Could differences in sampling time or HMGB1 turnover explain this?

Answer: Thank you for your question. It is reasonable that HMGB1 did not leak into the bloodstream to a significant degree, as this was a localized injury. In particular, rabbits have few collateral blood vessels, so HMGB1 leakage from injured tissue after reperfusion is likely to be limited. The small number of animals used and the exploratory nature of the study also had an impact. This issue is discussed in detail in the Discussion and Limitations sections of the new manuscript. (Page 15 Line 428-451, Page 18 Line 607- 618)

3.PCR Data Handling and Exclusion of Outliers

Several outliers in the qPCR dataset were excluded using IQR criteria. Given the small sample size, such exclusions may bias the results. Were these exclusions pre-specified? A sensitivity analysis including all values should be considered or at least discussed.

Answer:

We appreciate the suggestion. Given the small sample size and the skewed nature of qPCR data, we adopted the exact Mann–Whitney U test as our primary analysis, which does not assume normality and is robust to outliers. We additionally report effect sizes (Cliff’s δ with 95% CIs) and Hodges–Lehmann median differences. To address the reviewer’s concern about potential bias from outlier handling, we present sensitivity analyses including (i) results after excluding IQR-flagged values and (ii) Welch’s t-tests on log2-transformed data, as well as permutation p-values. The overall directional conclusions were preserved across approaches, indicating that our findings do not hinge on the choice of test or outlier handling.

In accordance with the changes, we have revised all sections related to Figure 10, including Method, Results, Discussion, and Conclusion. As these are quite extensive and fundamental changes, we will refrain from listing them here. Please refer to the main text. (Page 13 Line 348 – 359, Page 16 Line 501-519)

(Figure 10) We have made completely new Figure 10.

(Supplementals)   We have made Table 4S, 5S, and Figure 1S for Figure 10.

4.Lack of Long-Term Functional Outcome Assessment

The study endpoint was 48 hours post-reperfusion. Although the authors justify this based on previous literature, a longer follow-up (e.g., 7–14 days) would better represent delayed neuronal loss and functional recovery in translational terms.

Answer: Thank you very much for your comment. I have added other reviewers to the discussion regarding your comments. I have also added a note regarding the need for reconsideration before practical application. We have added some comments in the Discussion section. (Page 18 Line 582-612)

5.Gender Bias

Only male rabbits were used. While this is common in preclinical studies, the exclusion of females should be acknowledged as a limitation, particularly when studying neuroinflammatory processes that may be sex-dependent.

Answer: Thank you for your comment. I have added a note about gender bias to the Limitations section. (Page 19 Line 614-618)

6.Lack of Mechanistic Insight Beyond TLR4

The study focuses largely on HMGB1-TLR4 signaling. However, other HMGB1 receptors (e.g., RAGE) are known to contribute to I/R injury. Inclusion of additional markers or even speculative discussion of these alternative pathways would enhance the mechanistic depth.

Answer: Thank you for your valuable comments. This is exploratory research prior to practical application, and we believe that further research is needed to elucidate the mechanism. We have added a significant amount of information to the Discussion section on the mechanism. We hope you will take the time to read the Discussion section, which has been completely revised. (Page 14 Line 374 – Page 15 Line 416)

8.Figure Legends Are Too Brief

The figure legends are generally too short and lack detail. Each legend should fully describe what is shown, including the biological context, experimental groups, number of replicates, and statistical annotations. A single-sentence legend is insufficient.

Answer: Thank you for pointing that out. I have added a figure legend.

9.Typographical Error in Figure 4

There is a typographical error in the Y-axis label of Figure 4 (“motor Eurons”). This should be corrected to “motor neurons.”

Answer: Thank you very much for your comment. We have fixed this problem in Figure 4.

10.Figure 7 – Missing Representative Western Blot Images

Figure 7 presents quantification graphs for Western blot results and refers readers to raw data in the supplementary file, which is commendable. However, representative, well-labeled Western blot images should also be included in the main figure panel alongside the graphs to improve readability and transparency.

Answer: Thank you for your advice. I have added a representative image to Figure 7.

11.Language and Style

The English language requires improvement throughout the manuscript. Although generally understandable, the text contains grammatical errors, awkward phrasing, and overly long sentences. Professional editing is strongly recommended to improve clarity, flow, and readability.

Answer: Thank you for your comment. The English proofreading was done by Editage.

Round 2

Reviewer 1 Report

Comments and Suggestions for Authors

The authors have made an effort to respond to all of the reviewer's suggestions and have included these changes in the final version of the manuscript.

Reviewer 2 Report

Comments and Suggestions for Authors

ok